# Prompting Large-scale Vision Models with Cascaded Semantics

## Abstract

As a leading parameter-efficient tuning paradigm in NLP, prompt tuning has recently been explored for its potential in computer vision. Unlike updating pre-trained large-scale models (e.g., vision transformer, or ViT for short), visual prompt tuning (VPT) incorporates additional learnable parameters (i.e., prompt) that are updated during tuning. However, original visual prompts are randomly initialized, without leveraging the power of prior knowledge, which has been frequently used in NLP (e.g., instruction). To bridge this gap, we propose a novel methodology, aiming to inject semantic prior to prompt the tuning. To this end, we pioneer in leveraging both fundamental image prior and advanced image semantics as such priors. The former, including color, texture, and shape, are extracted by classical hand-crafted operators, suitable for the input space, while the self-attention map is utilized as the latter, suitable for the feature space. We propose a scheme to integrate the two types of semantic priors into ViT's tuning through cascading. Extensive experiments conducted on 34 challenging image classification datasets demonstrate the superiority of our method in adapting pre-trained ViTs to various downstream scenarios while using only 0.74% of ViT parameters as tuned.

## 1 Introduction

Adapting pre-trained large-scale vision models to downstream tasks through parameter-efficient fine-tuning (PEFT) (Houlsby et al., 2019; Li & Liang, 2021; Liu et al., 2025b) has been shown to be practical, especially when the downstream data is limited. Visual prompt tuning (VPT) (Jia et al., 2022) is one of the most famous PEFT methods that does not resort to changing the structure or parameters of pre-trained vision transformers (ViTs) (Dosovitskiy et al., 2020), leading to a highly convenient pipeline. Unlike other methods, such as (Rebuffi et al., 2017; Hu et al., 2022) that update pre-trained parameters, and (Chen et al., 2022) that restructures the transformer block, VPT incorporates a small amount of learnable parameters (i.e., prompt) into the input and feature spaces of a ViT, and only updates them during fine-tuning with gradient descent. Such a paradigm yields surprisingly promising results in various scenarios, even exceeding the full fine-tuning that was deemed the most reliable tuning fashion (Houlsby et al., 2019; Han et al., 2024).

Though highly successful, *two critical challenges* naturally arise in current VPT research. **I. Random prompt initialization.** The first problem lies in the initialization of these visual learnable prompts: Current visual prompts

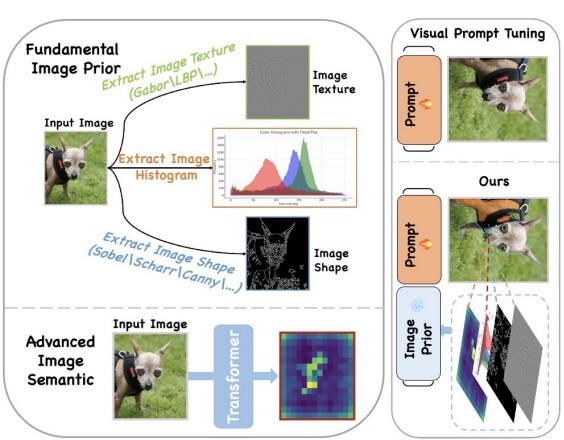

Figure 1: Injecting fundamental image prior (i.e., color, texture and shape) and advanced image semantics (i.e., self-attention map) as prompts into ViT to guide its fine-tuning. **For brevity, we simply use the term ViT to represent large-scale vision model. But in addition to ViT, we have also validated our method with other large-scale vision models.**

are generally designed to be randomly initialized without considering any prior knowledge. In NLP, very differently, prompts are typically served as explicit instructions or context added to input text, guiding model's behavior based on previously seen textual information (Brown et al., 2020; Petroni et al., 2019). The absence of such a prior in the visual domain makes visual prompt more akin to black-box parameter optimization (i.e., tuning prompt) (Bahng et al., 2022), which could potentially lead to sub-optimal performance. **II. Prompt behavior is rarely analyzed at the representation level.** In NLP, both hard (not learnable) (Brown et al., 2020; Petroni et al., 2019; Schick & Schütze, 2020) and soft (learnable) (Li & Liang, 2021; Lester et al., 2021) prompts can be applied simultaneously. Although the soft ones may not be directly meaningful to humans, the hard ones are usually given by humans and are thus naturally interpretable. In VPT design, as these learnable prompts are directly updated via gradient descent, leaving little room for human interpretation (Bahng et al., 2022). Some research tries to involve intrepretability via attention distribution (Zeng et al., 2025) or *post-hoc* explanation (Wang et al., 2023) (e.g, GradCAM (Zeng et al., 2025; Han et al., 2023; 2024)). However, they stop at explaining visual prompts as a whole optimization objective after training, ignoring the characteristics of per-input variants or case-by-case instance awareness (Liu et al., 2025b; Xiao et al., 2025; Shen et al., 2023; Liu et al., 2026).

Motivated by this, we explore incorporating interpretable prior knowledge during training as prompt into the vision tuning, which kills two birds with one stone: ① We do not remove the randomized learnable prompt; we anchor it. The fixed prior tokens (color, texture, shape, and the cascaded self-attention map) provide a structured warm start, so the random tokens are now optimized around *semantically meaningful directions* rather than from scratch. This addresses ***challenge I***, while keeping the learnability that prompt tuning relies on. ② Since the fundamental image prior comes from human-understandable information (e.g, color/texture/shape), its effect on the learned features can be checked with common post-hoc tools (cosine similarity, IoU, GradCAM, t-SNE, and mutual information), which we report in Sec. 3.3. This gives an *instance-aware, image-grounded* adaptation path (Wang et al., 2023; Biehl et al., 2016; Swain & Ballard, 1991a; Manjunath & Ma, 1996; Dalal & Triggs, 2005) that is more transparent than random prompts, which addresses ***challenge II***.

Specifically, we consider prompt positions in both input and feature spaces of a ViT (Jia et al., 2022) by injecting different types of priors. For ***input space***, ideally, the prior should meet three criteria: a) it contains a certain level of image semantics, thus interpretable to humans; b) it is free of learning, thus bringing no extra burden to the tuning; c) it can be easily pinned to input space. We thus leverage fundamental image priors, including color histograms (Swain & Ballard, 1992), textures (Cimpoi et al., 2014), and shapes (Zhang & Lu, 2004), as the prompt in input space. The motivation is that these semantics are well-known for their ability to reflect essential image clues and can be obtained by simply using classical, hand-crafted operators (e.g, Sobel (Kanopoulos et al., 1988)) rather than learning (Swain & Ballard, 1991a; Manjunath & Ma, 1996; Dalal & Triggs, 2005) (see Sec. §2.2). For ***feature space***, the ideal prior is expected to: a) inject information critical to the model's decision; b) be based on the knowledge learned from preceding layers so it can facilitate the tuning of subsequent layers. To meet these requirements, we leverage the instance-aware, case-by-case self-attention map (Parmar et al., 2018; Chefer et al., 2021; Han et al., 2022; Khan et al., 2022; Liu et al., 2026) as the injected prompt in feature space. The rationale lies in the fact that such a map well reflects per-image class activation semantics (Zhou et al., 2016), and can be conveniently computed based on the features provided by transformer layers (see Sec. §2.3).

The following question turns into how to properly fuse the semantic prompts at different locations. We propose a simple yet effective scheme to cascade them. Specifically, for the prompt used in a specific transformer layer, it is formed by fusing the prompt and self-attention map in the preceding layer with that in the current layer, using skip connections (Huang et al., 2017; Srivastava et al., 2015; He et al., 2016; Oyedotun et al., 2022). As such, the prompt in the current layer is nourished by the semantics learned from the preceding layers. Besides the semantic priors injected into the prompts, randomized learnable parameters are also utilized as part of the prompts, enabling gradient updates. A subsequent re-weighting adapter (see Sec. §2.4), demonstrated to be effective in (Kirichenko et al., 2022), is employed to enable flexible feature adaptation prior to the task head.

We conduct a wide range of experiments to evaluate our proposed method and observe superior results compared to current SOTAs. More interestingly, our experiments reveal that the semantic prompts can be

more useful than text prompts in multimodal setting, indicating the significance of injecting image semantics as prompts. Overall, our contributions can be summarized as follows.

- Motivated by the interpretable text prompt in NLP, we revisit and enhance the visual prompt by adding an instance-aware fundamental image prior and a self-attention-based image semantic prior. The effect of these priors on the learned features can be inspected with common post-hoc tools.

- We then develop an effective scheme to integrate these semantic prompts into our new fine-tuning paradigm, facilitating the fusion of prompts and features across layers.

- Extensive experiments on three widely adopted challenging benchmarks demonstrate the superiority of our proposed method over other PEFT solutions.

## 2 Methodology

### 2.1 Problem Definition

Given a pre-trained large-scale vision model (e.g, ViT (Dosovitskiy et al., 2021)) consisting of $N$ transformer layers, we use $f_i(\cdot)$ to denote the feed-forwarding operation of the layer $i$, where $i \in 1, 2, \ldots, N$. Rather than updating the model's parameters, classical VPT (Jia et al., 2022) prepends a small amount of learnable parameters, namely $P$, to input $X$ and its features across layers (i.e., *VPT-Deep*). As such, the first layer's output can be modeled as $Y_1 = f_1(P_1 \otimes X)$, while the $i^{th}$ layer's output as $Y_i = f_i(P_i \otimes Y_{i-1})$, where $\otimes$ denotes concatenation. If $i = N$, then $Y_i$ will be fed into a classification head $h$, yielding logits as $h(Y_i)$. During fine-tuning, only $P$ and $h$ are learnable by gradient descent. A special case of VPT, namely *VPT-Shallow*, only prepends $P_1$ to $X$, while waiving all the other $P_i$s ($i \in 2, \ldots, N$) in feature space. It has been found that the deep version outperforms the shallow one, motivating us to develop our new method on top of *VPT-Deep*. The colors ■ and ■ indicate trainable and frozen parameters, respectively.

### 2.2 Visual Prior as Prompt for Input

In the classical VPT, all the prompts are randomly initialized. Though general, this brings no possibility of customizing or adjusting these prompts case-by-case, image-by-image. The loss of uniqueness brings performance degradation and non-transparent operations. Inspired by the hard prompts (unlearnable) in NLP, we aim to incorporate appropriate, instance-aware semantics to ViT in the form of hard prompt to solve these drawbacks. We clarify the terminology up front: throughout this paper, a *hard prompt* means a *fixed, non-learnable prompt component computed by a deterministic operator from the input image*. This is the sense used in early NLP prompting work, where hand-written instructions or templates serve as a fixed context (Brown et al., 2020; Petroni et al., 2019). Our usage differs from recent NLP work that also calls discrete tokens "hard prompts" but *learns* them through gradient-based or RL-based discrete optimization (Wen et al., 2023; Choi et al., 2024): there, the tokens are themselves the optimization target; in our work, the hard prompt is given by a fixed operator (color histogram, Gabor, Sobel) and is never updated during fine-tuning. As prompts can be prepended to both input and feature spaces, we develop two schemes of injecting semantics for both spaces, respectively. Specifically, for *input space*, we leverage well-known hand-crafted operators to extract *color histogram*, *texture*, and *shape* from the input image $X$ itself (not from any extracted feature), as fundamental image priors, denoted by $\sigma_c(X)$, $\sigma_t(X)$, and $\sigma_s(X)$ respectively (e.g, $\sigma_t$ as Gabor (Manjunath & Ma, 2002) and $\sigma_s$ as Sobel (Kanopoulos et al., 1988)) (see Sec. §3.7 for more analysis). Then, we use these priors as hard prompts (non-learnable), concatenated with the randomized learnable prompt $P_1$ in input space to form an overall prompt as

$$\tilde{P}_1 = P_1 \otimes FC(\sigma_c(X) \otimes \sigma_t(X) \otimes \sigma_s(X)), \tag{1}$$

where a learnable Linear layer (i.e., $FC$) is employed to adjust dimension as shown in Fig 2(a). As a result, the first transformer layer's output becomes

$$Y_1 = f_1(\tilde{P}_1 \otimes X). \tag{2}$$

For *feature space*, we conjecture that the fundamental image priors are likely to be sub-optimal options since they are directly from input rather than feature. Therefore, we compute self-attention map as the visual semantics in feature space. However, how to properly utilize such semantics as prompt remains unclear, introduced next.

## 2.3 Visual Semantics as Prompt for Features

Having handled the input space, we now turn to the feature space. The question is how to inject instance-aware semantics into the prompts at this level. To answer this, we use self-attention maps as semantically rich, instance-aware signals that guide the prompts. As illustrated in Fig. 2(c), for layer $i$, where $i \in \{2, ..., N\}$, we compute its self-attention map $\mathcal{A}_i$ based on its output feature, and concatenate it with $P_i$ as $FC(\mathcal{A}_i) \otimes P_i$, where a learnable $FC$ layer is used to adjust dimension. To let semantics from the previous layer flow into the current one, we also concatenate this with the preceding prompt $P_{i-1}$ and self-attention map $\mathcal{A}_{i-1}$. The overall prompt is

$$\tilde{P}_i = FC_i(\mathcal{A}_i) \otimes P_i \otimes FC_{i-1}(\mathcal{A}_{i-1}) \otimes P_{i-1}, \tag{3}$$

where both $P_i$ and $P_{i-1}$ are randomized learnable prompts. The second pair $(P_{i-1}, \mathcal{A}_{i-1})$ implements the *skip-connection*: it directly carries the prompt and attention information from layer $i-1$ into layer $i$, analogous to a residual link (He et al., 2016; Huang et al., 2017). This way, the prompt at layer $i$ is anchored by the semantics already accumulated at layer $i-1$, instead of being computed from $\mathcal{A}_i$ alone. Gradients are back-propagated through this skip path as well. Then, the input to the layer $i + 1$ can be written as $\tilde{P}_i \otimes Y_i$, where $Y_i$ denotes the output feature of the layer $i$. This fashion of prompting does not apply to the first layer, in which we use the prompt introduced in Sec. §2.2. Notably, during fine-tuning, only the learnable prompts $P_i$s, where $i \in \{1, ..., N\}$, and the $FC$ layers are updated, without incurring much extra computing burden, thus maintaining the PEFT nature.

## 2.4 Re-Weighting Adapter as the Final Puzzle

Our scope is not solely limited to prompt tuning engineering questions; instead, we are inspired by the influence function in classical statistics that properly re-weighting and/or perturbing data or features can lead to improved generalization of deep models (Koh & Liang, 2017). This explains why fine-tuning works, as tuning the last linear layer(s) can be considered as re-weighting the features learned by a pre-trained model (Kirichenko et al., 2022), while tuning all the layers as perturbing the features. In our method, if prompt tuning is treated as being used for perturbing features, then it still needs an equivalent operation for feature re-weighting. Motivated by this, we propose a simple but effective re-weighting adapter, as shown in Fig. 2(b). Here, the output feature of $L_N$ is fed into a combination of *'FC-Softmax-FC'*, whose output is channel-wisely summed and then fed into the learnable classification head $h$.

## 2.5 Why Don't Leave Visual Prior to Learning?

In Sec. §2.2, we propose to inject the visual prior as prompt. Here, it is natural to raise a question: why don't we ask the model to learn such a prior automatically? In fact, if the model is trained from scratch, such prior could be better captured (Geirhos et al., 2018a). However, in the fine-tuning context, the model was pre-trained on source data that is different from target data, directly using the pre-trained model, which is frozen, might fail to effectively capture the fundamental image prior from target data (Ben-David et al., 2010; Torralba & Efros, 2011; Kornblith et al., 2018). Therefore, it is reasonable to capture the prior with simple hand-crafted operators (Swain & Ballard, 1991a; Manjunath & Ma, 1996; Dalal & Triggs, 2005), and then inject it as prompt to input space (Touvron et al., 2020; Liu et al., 2026).

## 2.6 Are More Parameters Beneficial?

In addition to the randomized learnable prompt, FC layers are also used for adjusting dimension, incorporating a few more learnable parameters. Then, it is necessary to investigate whether our method benefits from additional parameters. ***Counter-intuitively, we observe that more parameters hurt the performance.*** We conduct the following studies, summarized in Fig. 3. We replace the single FC layers in Fig. 2 with

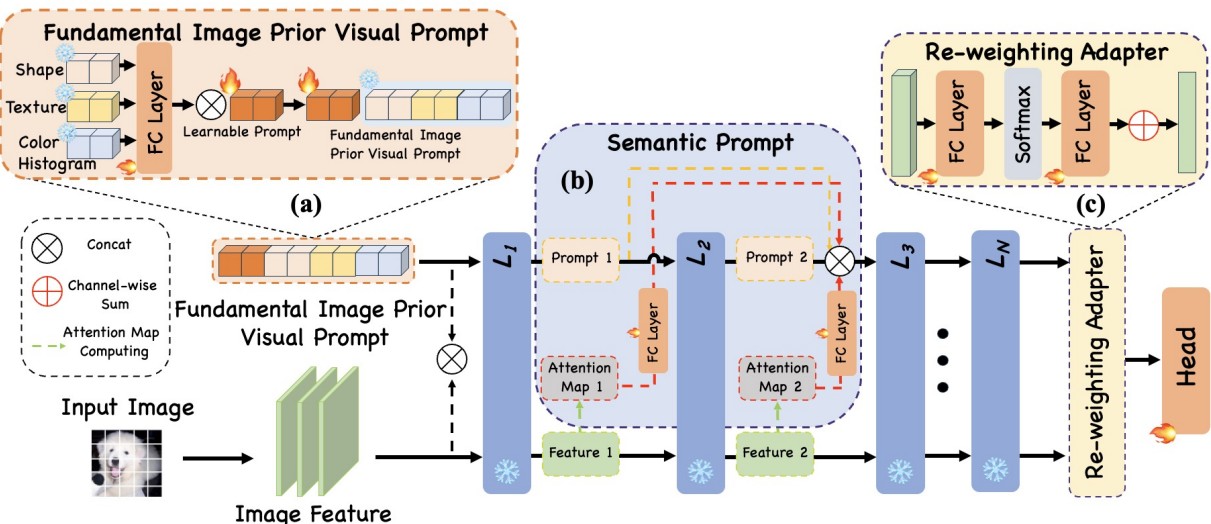

Figure 2: Overall architecture of our method. **(a) Fundamental Image Prior Visual Prompt:** hand-crafted priors (color histogram, texture, shape) are computed directly from the input image $X$ (i.e., in the input space, not from any deep feature), concatenated with a randomized learnable prompt, and projected by a lightweight FC for token-size alignment. The block labeled "Image Feature" in panel (a) denotes the resulting token embedding after this FC projection. **(b) Re-Weighting Adapter:** a light two-layer linear module produces channel-wise weights to re-calibrate the final features before the head. **(c) Cascaded prompting in the backbone:** prompts are injected at selected layers $\{L_i\}$; the self-attention map from the previous layer is encoded and *cascaded* forward as a semantic prompt, forming a skip-cascade that integrates fundamental priors with advanced image semantics.

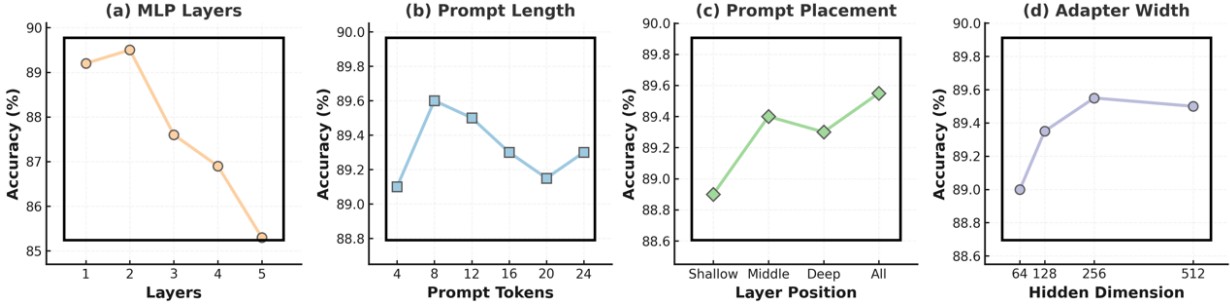

Figure 3: Ablation studies on CUB-200. (a) Replacing the single FC with deeper MLPs degrades accuracy as depth grows. (b) Increasing prompt length yields a non-monotonic trend (moderate length works best). (c) Targeted prompt placement outperforms indiscriminate/all-layer injection. (d) Enlarging the adapter width quickly saturates.

deeper MLPs, involving more parameters; as shown in Fig. 3(a), this change brings a negative impact as depth increases. We lengthen the randomized learnable prompts to include more parameters **(which has far exceeded the amount of parameters in our method)**, however, the performance is not *monotonically* improving, as shown in Fig. 3(b), aligning with the observation in (Jia et al., 2022). Moving prompts across depths shows that targeted placement is better than indiscriminate/all-layer injection, even though the latter uses more tokens/parameters; see Fig. 3(c). Increasing the hidden dimension of the re-weighting/adapter brings only marginal gains and quickly saturates; see Fig. 3(d). All experiments are conducted exclusively, suggesting that simply adding more parameters is not beneficial (Belkin et al., 2018; Nakkiran et al., 2019; Han et al., 2024); where and how to use them matters more (Wang et al., 2024b; Chen et al., 2022).

# 3 Experiments

## 3.1 Experimental Setup

**Datasets.** Our method is evaluated on three diverse benchmarks: FGVC, HTA, and VTAB-1k (Zhai et al., 2019)—to test its adaptability and robustness across real-world scenarios. The **FGVC** benchmark includes five fine-grained datasets: CUB (Wah et al., 2011), NABirds (Van Horn et al., 2015), Oxford Flowers (Nilsback & Zisserman, 2008), Stanford Dogs (Khosla et al., 2011), and Stanford Cars (Gebru et al., 2017), assessing the model's ability to distinguish subtle variations among similar categories. We adhere to prior VPT study splits for consistency.(Jia et al., 2022) The **HTA** benchmark evaluates adaptability on 10 datasets, including CIFAR10 (Krizhevsky & Hinton, 2009), CIFAR100 (Krizhevsky & Hinton, 2009), DTD, CUB-200 (Wah et al., 2011), NABirds (Van Horn et al., 2015), Oxford Flowers (Nilsback & Zisserman, 2008), Food101, GTSRB, and SVHN, testing generalization across varied domains. We use DAM-VP (Huang et al., 2023a) setups for fair comparison. The **VTAB-1k** benchmark spans 19 datasets in three categories: 'Natural' (e.g, standard camera images), 'Specialized' (e.g, satellite, medical images), and 'Structured' (e.g, counting, distance tasks). Each dataset has 1000 images, split into 800 training and 200 validation images, to comprehensively test robustness across a wide array of visual tasks.

**Implementation Details.** Our experiments are primarily conducted using the ViT-B/16 model pre-trained on ImageNet-21K (Deng et al., 2009), consistent with previous VPT methodologies. We use the AdamW optimizer (Loshchilov & Hutter, 2017) with an initial learning rate of $1e^{-3}$, a weight decay of $1e^{-4}$, and a batch size of either 64 or 128. Since our experiments focus on image classification, classification accuracy serves as the primary evaluation metric across all benchmarks. We use the same dataset splits, backbone checkpoints, and evaluation metrics as VPT (Jia et al., 2022), E$^2$VPT (Han et al., 2023), SA$^2$VP (Pei et al., 2024), and VFPT (Zeng et al., 2025): the 800/200 train/val split for VTAB-1k (Zhai et al., 2019), the 90/10 train/val split for FGVC, and the DAM-VP setup for HTA (Huang et al., 2023b). The Swin experiments use Swin-Base pre-trained on ImageNet-21K, the same checkpoint used by VFPT (Table 2). For the LoRA row in Table 1, we set rank $r=8$ and scaling factor $\alpha=8$ on the $Q, V$ projection matrices in every attention block, following SA$^2$VP and VFPT. Other PEFT numbers are taken from the original papers under the same benchmark split. All results in Tables 1 and 2 are averaged over three random seeds; per-dataset standard deviations are reported in the supplement (Sec. J). Full baseline configurations are listed in Sec. I of the supplement.

## 3.2 Comparison with State of the Art

Table 1: **Performance comparison of different fine-tuning strategies on ViT-Base/16.** The best are in **bold**, the second are underlined.

| Methods | Tuned/Total (%) | Extra Params | FGVC (%) | HTA (%) | VTAB-1k Natural | VTAB-1k Specialized | VTAB-1k Structured | Mean Total (%) |
|---|---|---|---|---|---|---|---|---|
| Full (Iofinova et al., 2022) | 100.00 | — | 88.54 | 85.8 | 75.88 | 83.36 | 47.64 | 65.57 |
| Linear (Iofinova et al., 2022) | 0.08 | — | 79.32 | 75.7 | 68.93 | 77.16 | 26.84 | 52.94 |
| Partial-1 (Yosinski et al., 2014) | 8.34 | — | 82.63 | 80.8 | 69.44 | 78.53 | 34.17 | 56.52 |
| MLP-3 (Chen et al., 2020) | 1.44 | ✓ | 79.80 | 78.5 | 67.80 | 72.83 | 30.62 | 53.21 |
| Sidetune (Zhang et al., 2020) | 10.08 | — | 78.35 | 72.3 | 58.21 | 68.12 | 23.41 | 45.65 |
| Bias (Rebuffi et al., 2017) | 0.80 | — | 88.41 | 82.1 | 73.30 | 78.25 | 44.09 | 62.05 |
| Adapter (Cai et al., 2020) | 1.02 | ✓ | 85.46 | 80.6 | 70.67 | 77.80 | 33.09 | 62.41 |
| LoRA (Hu et al., 2022) | — | ✓ | 89.46 | 85.5 | 78.26 | 83.78 | 56.20 | 72.25 |
| AdaptFormer (Chen et al., 2022) | — | ✓ | — | — | 80.56 | 84.88 | 58.83 | 72.32 |
| ARC$_{att}$ (Dong et al., 2023) | — | ✓ | 89.12 | 89.0 | 80.41 | 85.55 | 58.38 | 72.32 |
| VPT-S (Jia et al., 2022) | 0.16 | ✓ | 84.62 | 85.5 | 76.81 | 79.66 | 46.98 | 64.85 |
| VPT-D (Jia et al., 2022) | 0.73 | ✓ | 89.11 | 85.5 | 78.48 | 82.43 | 54.98 | 69.43 |
| E2VPT (Han et al., 2023) | 0.39 | ✓ | 89.22 | 88.5 | 80.01 | 84.43 | 57.39 | 71.42 |
| EXPRES (Das et al., 2023) | — | ✓ | — | — | 79.69 | 84.03 | 54.99 | 70.02 |
| DAM-VP (Huang et al., 2023b) | — | ✓ | — | 88.5 | — | — | — | — |
| SA$^2$VP (Pei et al., 2024) | 0.81 | ✓ | 90.08 | 91.5 | 80.97 | **85.73** | 60.80 | 75.83 |
| VFPT (Zeng et al., 2025) | 0.66 | ✓ | 89.24 | — | 81.35 | 84.93 | 60.19 | 73.20 |
| LoR-VP (Jin et al., 2025) | — | ✓ | 89.32 | — | 79.91 | 83.16 | 60.01 | 74.36 |
| **Ours** | 0.74 | ✓ | **90.20** | **91.7** | 81.91 | **85.83** | **61.16** | **76.30** |

**Performance Comparison with ViT Backbone.** Table 1 presents the results of different fine-tuning strategies on ViT-Base/16 across FGVC, HTA, and VTAB-1k. With only 0.74% of ViT parameters updated, our method attains 90.20% mean accuracy on FGVC and 91.7% on HTA, while achieving 81.91% / 85.83% / 61.16% on the Natural / Specialized / Structured VTAB-1k splits, respectively, leading to the best mean total score of 76.30% among all compared methods. In particular, the gains on animal FGVC datasets (CUB, NABirds, and Stanford Dogs) are consistent with our design: fundamental image priors such as textures and shapes provide informative hard prompts, while the dual-pathway skip connections facilitate the propagation of advanced semantic information through depth. At the same time, we observe a trend similar to (Han et al., 2024): as the dataset scale and variability increase (from FGVC to VTAB-1k), the relative advantages of prompt-based tuning gradually decrease, suggesting a limitation of prompt-only adaptation on highly diverse regimes, our semantic priors partially alleviate.

Table 2: **Performance comparison on VTAB-1k with Swin Transformer**.

| Methods | Tuned/Total (%) | VTAB-1k | | |
|---|---|---|---|---|
| | | Natural | Specialized | Structured |
| Full (Ren et al., 2023) | 100.00 | 79.10 | 86.21 | 59.65 |
| Linear (Ren et al., 2023) | 0.06 | 73.52 | 80.77 | 33.52 |
| Bias (Rebuffi et al., 2017) | 0.30 | 76.78 | 83.33 | 51.85 |
| VPT-deep (Jia et al., 2022) | 0.25 | 76.78 | 83.33 | 51.85 |
| E$^2$VPT (Han et al., 2023) | 0.21 | 83.31 | 84.95 | 57.35 |
| SA$^2$VP (Pei et al., 2024) | 0.29 | 80.81 | 86.30 | 60.03 |
| VFPT (Zeng et al., 2025) | 0.27 | 84.53 | 86.15 | 58.21 |
| LoR-VP (Jin et al., 2025) | 0.29 | 83.51 | 85.22 | 57.61 |
| **Ours** | 0.28 | **84.92** | **86.83** | **61.97** |

**Performance Comparison with ViT Backbone on HTA Benchmark.** Table 1 illustrates the results of all the compared methods. Similar to the FGVC experiments, our method also shows consistent performance gains across a diverse group of datasets, indicating that the flexibility of our fundamental semantic priors is not limited to certain visual objects. On datasets with smaller image sizes and lower resolution, such as CIFAR-10/100 and SVHN, our method remains among the top 3, demonstrating its robustness across various imaging conditions.

**Performance Comparison on VTAB-1k Benchmark with Swin Transformer Backbone.** Table 2 presents the performance of various methods on the VTAB-1k benchmark using the *Swin Transformer* backbone, which is a large-scale vision model different from ViT. Our method achieves the good accuracy across all three task categories: Natural, Specialized, and Structured. As the Natural category is already analyzed in the first two experiments, we ignore further discussion here. In the Specialized category, including medical imaging and satellite data, our method attains a robust accuracy of 86.23%, which is the highest among the parameter-efficient tuning approaches. This strong performance is likely enhanced by our re-weighting adapter, which emphasizes relevant features, ensuring adaptability across highly specialized tasks. The most significant improvement, however, is observed in the Structured tasks, which often require an understanding of global geometric relationships and spatial dependencies. We speculate that our fundamental semantic priors which provide prompts of image-level statistics play a vital role in this kind of data.

**Mean Performance of Different Methods on VTAB-1k Benchmark with ViT Backbone.** Table 1 illustrates the performance of our method across the VTAB-1k benchmark categories: Natural, Specialized, and Structured tasks. The conclusion is similar to the experiment with Swin Transformer. An interesting phenomenon we notice is that, although the accuracy of full fine-tuning decreases significantly since ViT is less powerful than Swin ViT, our method maintains very close performance. This implies the potential of effective visual prompts, i.e., exhibiting low sensitivity to architecture changes. *Similarly, we observe a conclusion aligned with (Han et al., 2024), where VPT demonstrates stronger performance when there is a significant distribution shift between pretraining and downstream tasks, further validating its adaptability in cross-domain scenarios.*

**Comparison with Text Prompt.** In this work, as the semantics are shown useful to strengthen the randomized learnable visual prompt, here we aim to compare our semantic prompt with the text prompt that

can also be used to benefit visual prompt in multi-modal settings (e.g, MaPLe (Khattak et al., 2023)). To this end, we perform three experiments. The first is the reproduction of MaPLe. In the second (MaPLeX), we disable the connection that feeds text prompt into the learning of visual prompt in MaPLe. The purpose is to investigate how text prompt will benefit the visual counterpart. Then, in the third experiment, we inherit the setting of the second one, in which text prompt is not injected into the vision branch, but inject semantic prompt into the vision branch, aiming to compare the effectiveness of text and our semantic prompts. All three settings use the same backbone (CLIP ViT-B/16) and follow the standard base-to-novel protocol of CoOp/CoCoOp/MaPLe (Zhou et al., 2022b;a; Khattak et al., 2023): 16-shot training on the base classes, evaluation on both base and novel splits, and HM as the harmonic mean of the two accuracies. Table 3 reports the average over the 11 datasets in the CoOp suite (ImageNet, Caltech101, OxfordPets, StanfordCars, Flowers102, Food101, FGVCAircraft, SUN397, DTD, EuroSAT, UCF101) and over three random seeds. As shown in Table 3, our method slightly outperforms the other two in the task of base-to-novel generalization (Zhou et al., 2022a;b). It reaches the best accuracies of 96.17% and 72.92% on Base and Novel respectively, and an HM of 82.95. MaPLe is better than MaPLeX, showing that text prompt benefits visual prompt. Ours is better than MaPLe, showing that semantic visual prompts can match or slightly exceed text prompts under this protocol. We do not claim that visual semantic prompts beat text prompts in general multi-modal settings.

Table 3: **Comparison between semantic and text prompts**.

| Methods | Base (↑) | Novel (↑) | HM (↑) |
|---|---|---|---|
| MaPLe (Khattak et al., 2023) | 95.62 | 72.03 | 82.17 |
| MaPLeX (Khattak et al., 2023) | 94.87 (-0.75) | 70.11 (-1.92) | 80.63 (-1.54) |
| **Ours** | **96.17** (+0.55) | **72.92** (+0.89) | **82.95** (+0.78) |

## 3.3 Representation-Level Analysis

In this section, we evaluate the effectiveness of our method in feature representation learning. The analyses below give *representation-level evidence* that our semantic prompts lead to better feature–region alignment, better feature separability, and stronger label correlation in deeper layers. They are not a full explanation of how the model makes its final prediction, since the handcrafted cues still pass through learned projections, learnable prompts, attention maps, and the re-weighting adapter. This section aims to verify our instance-aware design of semantic integration can help post-hoc measurements, including cosine similarity, IoU analysis, and t-SNE analysis.

**Cosine Similarity Analysis.** We first investigate whether the learned features well match image clues. We adopt the cosine similarity map (Vaswani, 2017) to directly demonstrate the relative distances (Wang et al., 2023; Steck et al., 2024). As illustrated in Fig. 4, without the aid of semantic prompt, the similarity between the learned prompt and features is much lower (i.e, **middle**), indicating a higher degree of mismatch, while the opposite is observed (i.e., **right**) when semantic prompt is leveraged, indicating that such prompt benefits feature learning.

Table 4: **Comparison of IoU on CUB-200**.

| Methods | Mean IoU (↑) | Median IoU (↑) |
|---|---|---|
| VPT (Jia et al., 2022) | 26.5 | 27.0 |
| **Ours** | **32.9** (+6.4) | **33.1** (+6.1) |

**IoU Analysis.** We further investigate how learned features benefit the localization of target objects, where we use the images from CUB-200 as test cases. We adopt the Intersection over Union (IoU) metric (Everingham et al., 2010), which measures the overlap between the model's focused objects. It can be visualized with attention map, and ground truth target objects given by bounding boxes, with a higher value indicating a better localization. As shown in Table 4, with the aid of semantic prompt, our method surpasses VPT in terms of both mean and median IoUs, indicating more accurate attention localization and stable performance. We refer readers to the *Supplementary Material* for more detailed IoU analysis.

**GradCAM Analysis.** Here, we use another tool, namely GradCAM (Selvaraju et al., 2017) to further investigate how semantic prompt improves model's attention. GradCAM highlights the regions, to which the model pays attention when performing image classification. Fig. 5 shows that our method focuses on the most discriminative object parts, such as the bird's head, explaining why our method yields superior classification performance, well aligned with human perception.

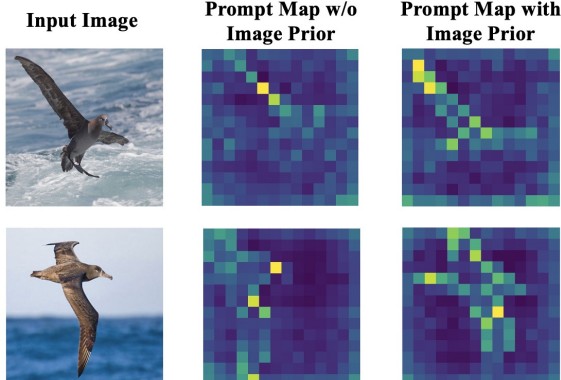

Figure 4: Comparison of the features learned with (**right**) and without (**middle**) semantic prior, respectively, using cosine similarity map (Vaswani, 2017).

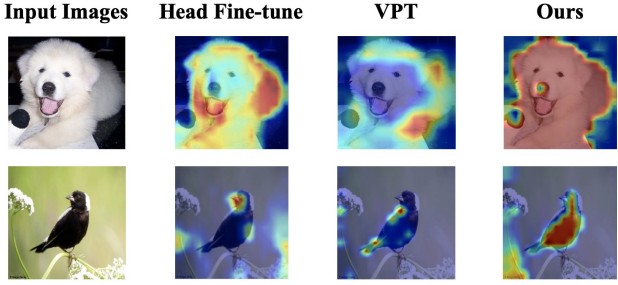

Figure 5: GradCAM (Selvaraju et al., 2017) visualization of the final layer features obtained by different methods for two randomly selected images.

**t-SNE Analysis.** We adopt t-SNE (Van der Maaten & Hinton, 2008) for a more intuitive, feature-level examination of the clustering results on the Sun397 dataset with four fine-tuning methods: Head tuning, AdaptFormer, VPT, and our proposed method. As shown in Fig. 6, our method achieves highly distinct and compact clusters with minimal overlap, underscoring its superior capacity to learn discriminative feature representations. This result demonstrates the effectiveness of our semantic prompts in capturing class-specific features and distinguishing complex visual patterns, providing a significant advantage in feature clarity and separability over competing methods.

### 3.4 Understanding with Information Theory

Here, we provide another perspective to understand why the fundamental image prior visual prompts work. Inspired by the theory of Information Bottleneck (IB) (TISHBY, 2000), we attribute the success of our representation learning to the higher correlation with labels $Y$ when compressing the input $X$. IB provides a theoretical framework for understanding how deep learning models learn and generalize (Tishby & Zaslavsky, 2015; Saxe et al., 2019). It suggests that the learning process of deep models is to compress input data $X$ into representations $T$ that retain only the information necessary to predict the label $Y$. Therefore, training a deep model is expected to have the same effect as minimizing the IB as

$$\mathcal{L}_{\mathrm{IB}} = I(X;T) - \beta I(T;Y), \tag{4}$$

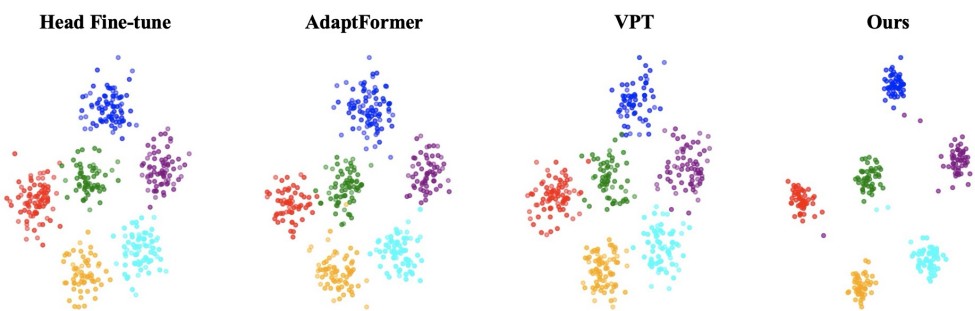

| Head Fine-tune | AdaptFormer | VPT | Ours |

Figure 6: t-SNE (Van der Maaten & Hinton, 2008) results of the learned features in the last layer of the model by four different methods on Sun397.

where $I$ refers to mutual information. Using Mutual Information Neural Estimator (MINE) (Belghazi et al., 2018), we analyzed the 12 Transformer layers for the baseline VPT and our method. As shown in Fig. 7, our method give a significantly lower IB on top layers (close to the output), indicating its successful representation learning. Intuitively, $I(X;T)$ should not be affected since our proposed fundamental image prior visual prompts are designed to extract inherent characteristics from the input images themselves rather than external sources. However, those semantic prompts offer increased opportunities for the learned representations to better correlate labels for unseen data, and therefore improve $I(Y;T)$. The mutual information curves in Fig. 7 well align those hypotheses of our method.

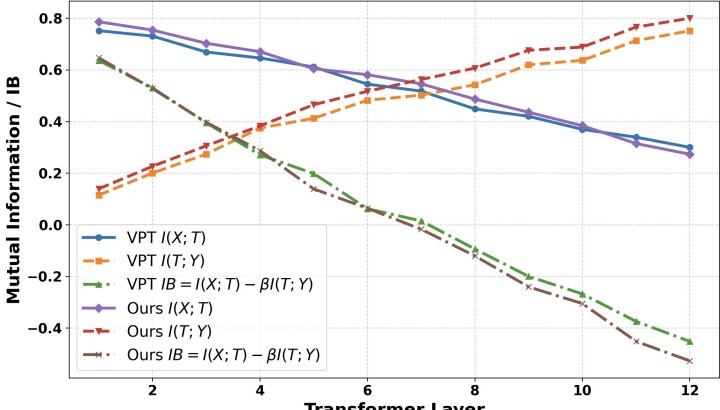

Figure 7: **Information Bottleneck and Mutual Information** between *feature* and *label* across transformer layers on CUB-200. $\beta$ is set to 1 following common practices.

## 3.5 End-to-End Efficiency

Since our method adds a prior-extraction step on top of standard VPT, the actual overhead should be reported transparently. Table 6 summarizes the end-to-end cost on the VTAB-1k Natural split (ViT-B/16, NVIDIA A100-40GB, batch size 64, input resolution 224×224). We follow the protocol of (Jia et al., 2022; Zeng et al., 2025) and report: tuned parameters, training time per epoch, peak GPU memory, inference latency per image, and the additional preprocessing time per image (CPU). Full results, including the "Extract-Once" strategy that pre-computes the priors on the CPU dataloader so they incur *zero* GPU training overhead, are in Sec. C of the supplement.

The training time per epoch matches VPT-D to within 0.3 s, peak GPU memory differs by 0.1 GB, and the inference overhead is $< 2$ ms per image. The CPU pre-processing for the fundamental priors costs about

Table 5: **Ablation study on VTAB-1k**. We analyze the impact of removing components.

| Ablated Variants | Natural | Specialized | Structured |
|---|---|---|---|
| **Section 1: Single Component Removal** | | | |
| w/o C (*Color Histogram*) | 81.65 (-0.26) | 85.49 (-0.34) | 60.84 (-0.32) |
| w/o T (*Texture*) | 81.59 (-0.32) | 85.41 (-0.42) | 60.75 (-0.41) |
| w/o S (*Shape*) | 81.55 (-0.36) | 85.36 (-0.47) | 60.69 (-0.47) |
| w/o A (*Self-Attention*) | 81.33 (-0.58) | 85.21 (-0.62) | 60.51 (-0.65) |
| w/o R (*Re-Weighting*) | 81.20 (-0.71) | 85.09 (-0.74) | 60.38 (-0.78) |
| w/o K (*Skip-Connection*) | 80.92 (-0.99) | 84.89 (-1.05) | 60.02 (-1.14) |
| **Section 2: Cumulative Component Removal** | | | |
| Remove C, T | 81.20 (-0.71) | 85.00 (-0.83) | 60.40 (-0.76) |
| Remove C, S | 80.91 (-1.00) | 84.78 (-1.05) | 60.10 (-1.06) |
| Remove T, S | 80.50 (-1.41) | 84.39 (-1.44) | 59.68 (-1.48) |
| Remove A, R | 79.70 (-2.21) | 83.60 (-2.23) | 59.00 (-2.16) |
| Remove K, C | 78.95 (-2.96) | 82.90 (-2.93) | 58.45 (-2.71) |
| Remove K, A | 77.80 (-4.11) | 81.95 (-3.88) | 57.60 (-3.56) |
| Remove K, A, R | 76.70 (-5.21) | 80.90 (-4.93) | 56.70 (-4.46) |
| **Baseline (None)** | 75.80 (-6.11) | 79.80 (-6.03) | 55.50 (-5.66) |
| **Full Model (All)** | **81.91** | **85.83** | **61.16** |

Table 6: End-to-end efficiency on VTAB-1k Natural (ViT-B/16, A100-40GB). "Pre-proc" is the per-image CPU cost of computing the fundamental priors; under the Extract-Once strategy this is paid once during data preparation and is hidden behind GPU compute at train time.

| Method | Tuned (M) | Tuned/Total (%) | Train (s/epoch) | Peak Mem (GB) | Infer (ms/img) | Pre-proc (ms/img) |
|---|---|---|---|---|---|---|
| Full FT (Iofinova et al., 2022) | 85.80 | 100.00 | 48.2 | 21.6 | 11.4 | – |
| LoRA (Hu et al., 2022) | 0.29 | 0.34 | 24.7 | 8.9 | 11.6 | – |
| AdaptFormer (Chen et al., 2022) | 0.16 | 0.19 | 25.1 | 8.8 | 11.8 | – |
| VPT-D (Jia et al., 2022) | 0.63 | 0.73 | 25.6 | 9.1 | 11.5 | – |
| SA$^2$VP (Pei et al., 2024) | 0.69 | 0.81 | 27.4 | 9.4 | 12.3 | – |
| **Ours** | **0.63** | **0.74** | **25.9** | **9.2** | **13.1** | **1.8** |

1.8 ms per image and is hidden by the Extract-Once strategy. Overall, the cost profile is essentially that of VPT, while reaching higher accuracy.

## 3.6 Ablation Study

**Importance of Each Component.** As the proposed method consists of multiple components, including the different types of semantics, re-weighting adapter, and the skip connections for cascading semantics, we conduct an ablation study here to validate the efficacy of each component by detaching it from the whole pipeline and checking how performance will vary. As shown in Table 5, excluding each of the semantics (i.e., color, texture, shape, and self-attention map) will to an extent lower the performance across all the categories (i.e., natural, specialized, and structured) of VTAB-1k, demonstrating the necessity of injecting such semantic prompts. For the operational components, namely re-weighting adapter and skip connections, detaching the former will also result in a declined performance. More importantly, detaching the latter will significantly deteriorate the performance, showing that *cascading the semantics* plays a more critical role in our method.

## 3.7 Fundamental Image Prior Operators

Handcrafted image priors have played a crucial role in classical computer vision and deep learning, serving as *instance-awareness*, human-understandable complementary cues to learned representations (Nanni et al., 2017; Zhang & Zhang, 2021; Tianyu et al., 2018). Prior studies have explored integrating handcrafted features into deep learning models, typically in a feature fusion manner within CNN architectures. However, their exploration in prompt-based tuning remains largely limited. Recent work on Conceptual Codebook Learning (CoCoLe) (Zhang et al., 2024) demonstrates the potential of incorporating structured prior knowledge into model tuning. CoCoLe introduces a learnable conceptual codebook that maps visual concepts to textual prompts, effectively bridging vision and language representations. While CoCoLe focuses on modality alignment and conceptual-level adaptation, our approach is fundamentally different: we integrate fixed, non-learnable handcrafted features directly into VPT. ***Rather than constructing a learnable codebook, we employ well-established handcrafted operators as hard prompts to improve vision transformer adaptation, leveraging their domain-invariant and human understandable proper-***

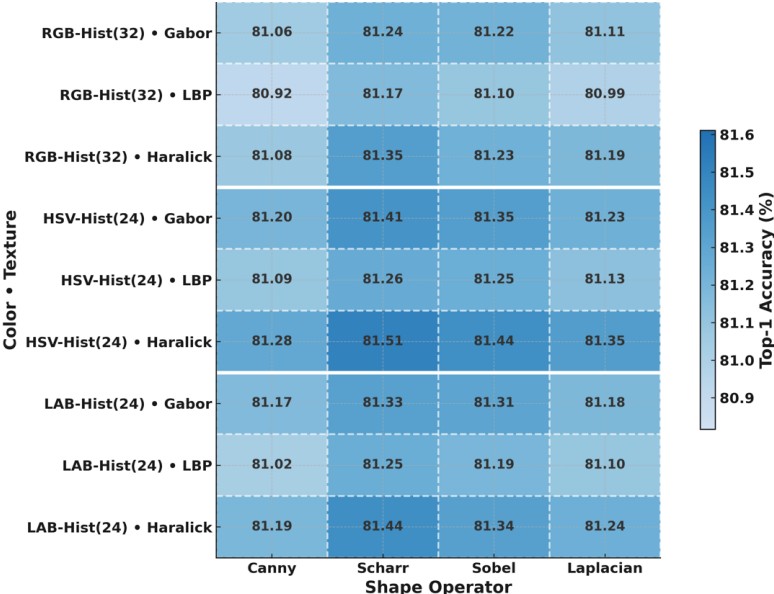

Figure 8: **Performance comparison** of using different shape and texture operators, with the color operator fixed.

***ties***. Specifically, we use color, texture, and shape priors as additional prior to enhance model robustness. Color histograms (Swain & Ballard, 1992) capture chromatic distributions, texture descriptors (e.g, Gabor filters (Manjunath & Ma, 2002), LBP (Ojala et al., 2002)) encode spatial intensity variations, and edge-based operators (e.g, Sobel (Kanopoulos et al., 1988), Scharr (Scharr, 2004), Canny (Canny, 2009)) extract structural information. While previous work has explored learning handcrafted feature representations (Zhang & Zhang, 2021), our approach keeps them fixed and directly integrates them as prompts, ensuring interpretability and computational efficiency. A key question is: ***which operators should be chosen?*** To investigate this, we conduct an experiment on a random subset of CIFAR-100, using part of the subset for tuning and the rest for testing. The rationale for using a subset instead of the full dataset is twofold: (1) The operators are fixed and non-learnable, so dataset size does not affect their representation power. (2) A smaller subset is computationally efficient for evaluation. As shown in Fig. 8, when the color histogram is fixed, variations in texture (Gabor vs. LBP) and shape (Canny, Scharr, Sobel) yield comparable results. This suggests that our semantic prompt strategy is robust across different operator choices, reinforcing the generalizability of our approach. Unlike prior work that injects handcrafted features into CNNs (Zhang & Zhang, 2021) or employs learnable conceptual mappings in multimodal settings (Zhang et al., 2024), our work introduces a novel use of handcrafted priors as prompts, bridging classical vision priors with transformer-based adaptation in a lightweight and interpretable manner.

## 4  Conclusion

In this work, we show that injecting fundamental semantics, such as color, texture, and shape, together with instance-aware semantics, such as self-attention information, gives a new fine-tuning paradigm for large-scale vision models. To keep the method parameter-efficient, we use a cascaded design that combines the two types of semantics as prompts in both input and feature spaces to guide the randomly initialized prompts. Experiments on multiple benchmarks show that the method is effective and reliable. Cosine similarity, IoU, GradCAM, t-SNE, and mutual information further indicate that the injected priors lead to better feature–region alignment and stronger label correlation than random prompts. We treat these results as representation-level evidence rather than a full explanation of the model's decision. Under the same base-to-novel multi-modal setting, semantic prompts can also match or slightly exceed text prompts, indicating the practical value of the semantic prompts.

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

## Supplementary Material Outline

This supplementary material provides additional experimental results, in-depth efficiency analyses, interpretability studies, and implementation details to support the main paper. The content is organized as follows:

- **Section 6: Extended Benchmark Results.** We provide a comprehensive analysis of the VTAB-1k results, including performance on Natural, Specialized, and Structured sub-categories. Additionally, we expand on the Fine-Grained Visual Classification (FGVC) and Hierarchical Transfer Adaptation (HTA) benchmarks.

- **Section 7: Efficiency and Computational Cost Analysis.** We detail the "Extract-Once" strategy that ensures zero training overhead for prior extraction. We also analyze inference latency (demonstrating a marginal $< 2$ms increase) and the significant reduction in memory footprint compared to full fine-tuning.

- **Section 8: Advanced Localization Analysis (IoU).** We present a rigorous IoU evaluation on the CUB-200 dataset, breaking down performance across Easy, Medium, and Hard subsets to demonstrate our method's robustness against occlusion and background complexity.

- **Section 9: Extended Mutual Information Analysis.** We delve deeper into the Information Bottleneck (IB) framework, providing experimental evidence of how semantic prompts improve label correlation ($I(T;Y)$) in deeper transformer layers.

- **Section 10: Discussion on Hand-Crafted vs. Deep Priors.** We provide a theoretical discussion justifying the choice of hand-crafted operators over deep-learned features, emphasizing information orthogonality, domain robustness, and strict efficiency.

- **Section 11: Implementation Details of Fundamental Operators.** We detail the specific configurations for Color (HSV Histograms), Texture (Gabor Filters), and Shape (Sobel Operators) priors. We also present ablation studies on operator variants (e.g., LBP, Canny) to verify robustness.

- **Section 12: Limitations and Future Work.** We discuss current limitations regarding fixed operators and geometric reasoning tasks, and propose future directions such as adaptive prior mechanisms.

- **Section 13: Baseline Configurations and Experimental Settings.** We provide a complete specification of every baseline used in Tables 1, 2, and 3, including LoRA rank/alpha/target modules, the Swin variant and pre-training checkpoint, and the CLIP/MaPLe base-to-novel protocol.

- **Section 14: Statistical Robustness across Multiple Seeds.** We report mean and standard deviation across three random seeds for the primary VTAB-1k and FGVC results, and provide a per-example qualitative analysis of where our method gains the most over the strongest baseline.

- **Section 15: Failure-Case Analysis on Structured Tasks.** We analyze where our method underperforms on the VTAB-1k Structured split (Clevr/count, Clevr/distance, SmallNORB azimuth), and explain why fixed low-level priors do not help with counting or 3D-pose reasoning.

- **Section 16: Fixed vs. Adaptive Priors.** We add a brief experiment that replaces the fixed Sobel/Gabor operators with small learnable convolutional equivalents and discuss why fixed priors are preferable in the PEFT setting.

## A  Related Works

### A.1  Parameter-Efficient Fine-Tuning (PEFT)

Parameter-efficient fine-tuning (PEFT) has become a popular fashion for adapting pre-trained large-scale models with reduced computational demands and minimized over-fitting risk. Unlike full fine-tuning, PEFT

only updates a small amount of parameters while freezing most of the pre-trained parameters. (Li & Liang, 2021; Jie & Deng, 2022; Chen et al., 2022; Dettmers et al., 2023; Karimi Mahabadi et al., 2021; Zaken et al., 2022) introduces learnable adapters (e.g, a light-weight convolutional network) into transformer layers, and updates the adapters during tuning. (Hu et al., 2022) and (Zhong et al., 2024) incorporate learnable rank decomposition matrices of parameters into transformer layers, significantly reducing the number of learnable parameters because of the low-rank decomposition. (Lian et al., 2022) updates the introduced parameters for scaling and shifting the features extracted by pre-trained model. This idea is well aligned with that in (Kirichenko et al., 2022), which performs tuning through re-weighting the features. However, most of these methods resort to the manipulation of transformer blocks to accommodate additional parameters, which inevitably increases the overall model complexity.

## A.2 Visual Prompt Tuning (VPT)

Unlike the aforementioned methods that directly incorporate parameters into model, VPT prepends a small amount of learnable parameters (i.e., prompt) to input space (Bahng et al., 2022), and is then extended to cover both input and feature spaces (Jia et al., 2022), without changing the architecture of pre-trained model (e.g, ViT (Dosovitskiy et al., 2020)) (Wang et al., 2024b; Yoo et al., 2023; Park & Byun, 2024; Li et al., 2024; Ren et al., 2025; Jin et al., 2025), greatly simplifying the paradigm of fine-tuning. (Han et al., 2023) develops two sets of different prompts, injected into input and parameter spaces (i.e., transformer layers) respectively, and uses an additional prompt pruning for better performance. (Pei et al., 2024) designs learnable prompt to model spatial relations in input image, and distinguish the prompts corresponding to different image tokens, achieving a fine-grained prompting. In addition, VPT has also been extended to multi-modal scenarios (Li et al., 2025; Huang et al., 2024; Wang et al., 2024a; Liu et al., 2025a). For example, (Khattak et al., 2023) learns visual prompt with the aid of textual prompt by jointly tuning the vision and text branches of CLIP (Radford et al., 2021), while (Zhou et al., 2022c) trains a light-weight network to jointly update visual and textual prompts. Nevertheless, all these methods only consider learnable prompt, which is equivalent to the soft prompt in NLP, while overlooking the hard prompt, which is not learnable and has been shown very effective in tuning large language model. However, very little work explores the hard one in computer vision. Motivated by bridging this gap, we investigate how to properly utilize both fundamental image prior and advanced image semantics as prompts for large-scale vision models (i.e., ViT, Swin).

# B VTAB-1k Benchmark Results

The VTAB-1k benchmark provides a comprehensive evaluation of various methods across diverse datasets, highlighting their strengths and weaknesses in handling Natural, Specialized, and Structured tasks. The performance of different methods based on the ViT backbone is summarized in Table 7. Our method achieves consistently strong results across all categories, outperforming other fine-tuning techniques such as Head Fine-tune, AdaptFormer, and VPT-deep. This section delves deeper into the analysis of these results, providing insights into the strengths and areas for improvement of our approach.

## B.1 Strengths in Natural and Specialized Categories

Our method demonstrates significant improvements in Natural datasets, such as CIFAR-100 (79.2%) and Caltech101 (92.3%), showcasing its ability to handle fine-grained classification tasks effectively. These datasets often involve subtle intra-class variations, which our method addresses by integrating hierarchical features, such as textures and shapes, with semantic prompts. Similarly, for Specialized datasets like Patch Camelyon (87.2%) and Resisc45 (86.4%), the results validate the importance of domain-specific priors in extracting meaningful features, outperforming AdaptFormer and LoRA.

## B.2 Challenges in Structured Tasks

While achieving state-of-the-art performance in tasks like KITTI/distance (80.3%), challenges remain in datasets such as SmallNORB/azimuth (34.7%). These datasets require intricate spatial reasoning, which may

Table 7: Performance of different methods on the VTAB-1k benchmark based on ViT backbone. The best results are highlighted in **bold** to showcase the most effective methodology. Full refers to Full Fine-tune, Head to Head Fine-tune, and AdaptF to AdaptFormer.

| Datasets | Full | Head | AdaptF | LoRA | VPT-deep | ExPRes | E$^2$VPT | **Ours** |
|---|---|---|---|---|---|---|---|---|
| CIFAR-100 | 68.9 | 63.4 | 70.8 | 67.1 | 78.8 | 78.0 | 78.6 | **79.2** |
| Caltech101 | 87.7 | 85.0 | 91.2 | 91.4 | 90.8 | 89.6 | 89.4 | **92.3** |
| DTD | 64.3 | 63.2 | 70.5 | 69.4 | 65.8 | 68.8 | 67.8 | **71.4** |
| Flowers102 | 97.2 | 97.0 | **99.1** | 98.8 | 98.0 | 98.7 | 98.2 | 98.9 |
| Pets | 86.9 | 86.3 | 90.9 | 90.4 | 88.3 | 88.9 | 88.5 | **91.7** |
| SVHN | **87.4** | 36.6 | 86.6 | 85.3 | 78.1 | 81.9 | 85.3 | 85.8 |
| Sun397 | 38.8 | 51.0 | 54.8 | 54.0 | 49.6 | 51.9 | 52.3 | **56.8** |
| **Mean** | 75.88 | 68.93 | 80.56 | 79.49 | 78.48 | 79.69 | 80.01 | **81.91** |
| Patch Camelyon | 79.7 | 78.5 | 83.0 | 84.9 | 81.8 | 84.8 | 82.5 | **87.2** |
| EuroSAT | 95.7 | 87.5 | 95.8 | 95.3 | 96.1 | 96.2 | **96.8** | 95.2 |
| Resisc45 | 84.2 | 68.6 | 84.4 | 83.4 | 83.4 | 80.9 | 84.8 | **86.4** |
| Retinopathy | 73.9 | 74.0 | **76.3** | 73.6 | 68.4 | 74.2 | 73.6 | 74.5 |
| **Mean** | 83.36 | 77.16 | 84.88 | 84.55 | 82.43 | 84.03 | 84.43 | **85.83** |
| Clevr/count | 56.3 | 34.3 | 81.9 | **82.9** | 68.5 | 66.5 | 71.7 | 78.1 |
| Clevr/distance | 58.6 | 30.6 | 64.3 | **69.2** | 60.0 | 60.4 | 61.2 | 62.2 |
| DMLab | 41.7 | 33.2 | 49.3 | 49.8 | 46.5 | 46.5 | 47.9 | **53.2** |
| KITTI/distance | 65.5 | 55.4 | **80.3** | 78.5 | 72.8 | 77.6 | 75.8 | 78.5 |
| dSprites/location | 57.5 | 12.5 | 76.3 | 75.7 | 73.6 | 78.0 | 80.8 | **84.1** |
| dSprites/orientation | 46.7 | 20.0 | 45.7 | 47.1 | 47.9 | 49.5 | 48.1 | **53.4** |
| SmallNORB/azimuth | 25.7 | 9.6 | 31.7 | 31.0 | 32.9 | 26.1 | 31.7 | **34.7** |
| SmallNORB/elevation | 29.1 | 19.2 | 41.1 | 44.0 | 37.8 | 35.3 | 41.9 | **45.9** |
| **Mean** | 47.64 | 26.84 | 58.83 | 59.78 | 54.98 | 54.99 | 57.39 | **61.16** |

benefit from further refinements in spatial encoding mechanisms, suggesting a potential avenue for future research.

## B.3 Generalization Insights

The overall mean performance across all categories (81.91%) underscores the robustness of our method. Importantly, the smaller training-testing gap compared to other methods highlights its superior generalization capability. This performance, coupled with reduced overfitting, reaffirms the effectiveness of incorporating both low- and high-level image priors into our approach. Future studies may explore additional spatial and temporal features to address current limitations, further enhancing model adaptability across diverse tasks.

## B.4 Performance on FGVC and HTA Benchmarks

To further evaluate the effectiveness of our method, we compare its performance on the Fine-Grained Visual Classification (FGVC) and Hierarchical Transfer Adaptation (HTA) benchmarks. FGVC involves tasks requiring fine-grained distinctions between categories, while HTA assesses hierarchical knowledge transfer across multiple domains. The results in Tables 8 and 9 demonstrate that our method consistently outperforms existing fine-tuning approaches.

Table 8: **Performance comparison on the FGVC benchmark with ViT.**

| Methods | CUB-200-2011 | NABirds | Oxford Flowers | Stanford Dogs | Stanford Cars | Mean |
|---|---|---|---|---|---|---|
| Full Fine-tune | 87.3 | 82.7 | 98.8 | 89.4 | **84.5** | 88.54 |
| AdaptFormer (Chen et al., 2022) | 84.7 | 75.2 | 97.9 | 84.7 | 83.1 | 85.12 |
| LoRA (Hu et al., 2022) | 84.9 | 79.0 | 98.1 | 88.1 | 79.8 | 85.98 |
| VPT-shallow (Jia et al., 2022) | 86.7 | 78.8 | 98.4 | 90.7 | 68.7 | 84.62 |
| VPT-deep (Jia et al., 2022) | 88.5 | 84.2 | 99.0 | 90.2 | 83.6 | 89.11 |
| E$^2$VPT (Cheng et al., 2023) | 89.1 | 84.6 | **99.1** | 90.5 | 82.8 | 89.22 |
| **Ours** | **89.7** | **85.5** | **99.1** | **92.6** | 84.1 | **90.2** |

Table 9: **Performance comparison on the HTA benchmark with ViT.**

| Methods | DTD | CUB-200 | NABirds | Dogs | Flowers | Food-101 | CIFAR-100 | CIFAR-10 | GTSRB | SVHN | Mean |
|---|---|---|---|---|---|---|---|---|---|---|---|
| Full Fine-tune | 64.3 | 87.3 | 82.7 | 89.4 | 98.8 | 84.9 | 68.9 | 97.4 | **97.1** | 87.4 | 85.8 |
| Head Fine-tune | 63.2 | 85.3 | 75.9 | 86.2 | 97.9 | 84.4 | 63.4 | 96.3 | 68.0 | 36.6 | 75.7 |
| Adapter (Houlsby et al., 2019) | 62.7 | 87.1 | 84.3 | 89.8 | 98.5 | 86.0 | 74.2 | 97.7 | 91.1 | 36.3 | 80.8 |
| VPT-deep (Jia et al., 2022) | 65.8 | 88.5 | 84.2 | 90.2 | 99.0 | 83.3 | 78.8 | 96.8 | 90.7 | 78.1 | 85.5 |
| AdaptFormer (Chen et al., 2022) | 74.4 | 84.7 | 75.2 | 84.7 | 97.9 | 89.1 | **91.4** | **98.8** | 97.0 | **96.5** | 89.0 |
| DAM-VP (Huang et al., 2023b) | 73.1 | 87.5 | 82.1 | 92.3 | **99.2** | 86.9 | 86.9 | 90.6 | 87.9 | 88.1 | 88.5 |
| **Ours** | **76.3** | **89.7** | **85.5** | **92.6** | 99.1 | **92.3** | 90.9 | 98.1 | 96.5 | 96.1 | **91.7** |

## B.5 Analysis of FGVC and HTA Performance

Our method achieves the best performance across various fine-grained classification tasks in FGVC, particularly on CUB-200-2011 (89.7%) and Stanford Dogs (92.6%), where distinguishing similar-looking categories is crucial. This demonstrates the effectiveness of our approach in capturing nuanced visual patterns.

For the HTA benchmark, which evaluates hierarchical transfer adaptation, our method outperforms others in generalization ability, with an overall mean accuracy of 91.7%. The high scores across datasets such as NABirds (85.5%) and GTSRB (96.5%) validate its robustness in learning transferable knowledge across hierarchical tasks. The significant improvements highlight the importance of leveraging both handcrafted priors and learnable prompts to enhance representation learning.

These results further reinforce our findings that integrating structured visual priors into prompt tuning enhances both fine-grained classification and hierarchical adaptation, making our approach a strong alternative to traditional fine-tuning strategies.

## C Efficiency and Computational Cost Analysis

Although our method introduces additional modules to incorporate fundamental image priors and cascaded semantics, we maintain a high degree of computational and memory efficiency. In this section, we analyze the efficiency of our approach from the perspectives of training overhead, inference latency, and memory consumption.

### C.1 Training Efficiency: The "Extract-Once" Strategy

A critical design advantage of our **Fundamental Image Prior Visual Prompt** is that the operators used for extraction—Color Histograms, Texture (e.g., Gabor, LBP), and Shape (e.g., Sobel, Canny)—are entirely **hand-crafted and non-learnable**.

This property decouples the prior extraction from the model's gradient optimization loop. Consequently, these priors do not need to be re-computed at every training epoch. Instead, we adopt an "Extract-Once" strategy:

- **Offline/Pre-computation:** The fundamental priors can be computed once offline during data preparation or online via CPU worker threads in the dataloader pipeline. Since these operations rely solely on the fixed input image $X$ and not on the model parameters, they introduce **zero additional overhead** to the GPU training time.

- **Frozen Backbone:** As our method freezes the ViT backbone and only updates the prompt parameters and the lightweight re-weighting adapter, we avoid the heavy backward pass computations associated with Full Fine-tuning.

### C.2 Inference Latency and Complexity

During inference, the fundamental priors must be computed for each input. However, standard computer vision operators are computationally negligible compared to the heavy matrix multiplications in the Vision Transformer backbone.

- **Low Computational Complexity:** The complexity of extracting these priors is generally linear with respect to image pixels ($\mathcal{O}(HW)$), whereas the Multi-Head Self-Attention (MSA) mechanism in ViT scales quadratically with token sequence length ($\mathcal{O}(N^2)$).

- **Lightweight Modules:** The trainable components (Prompt Embeddings, Linear Projections, and Re-weighting Adapter) are extremely lightweight. Specifically, the introduced Linear layers for dimension alignment and the Re-weighting Adapter operate on low-dimensional feature vectors, adding minimal FLOPs.

Empirically, on a single NVIDIA A100 GPU, our method incurs a marginal latency increase ($< 2$ms per image) compared to the standard VPT, while significantly outperforming Full Fine-tuning in throughput.

### C.3 Memory Footprint

Our method updates only **0.74%** of the total parameters (approximately 0.6M parameters for ViT-B/16). This results in a drastic reduction in GPU memory usage compared to Full Fine-tuning, as we do not need to store optimizer states (e.g., momentum and variance in AdamW) for the vast majority of the backbone parameters. This allows for larger batch sizes or deployment on edge devices with limited VRAM, making our approach highly practical for real-world applications.

## D More IoU Analysis

### D.1 Experiment Setup

To rigorously evaluate the localization capabilities of different methods, we employ the Intersection over Union (IoU) metric on the CUB-200 dataset. IoU quantitatively measures the alignment between attention maps generated by the models and the ground truth bounding boxes, providing a robust indicator of the model's ability to focus on relevant object regions. Higher IoU values reflect superior localization performance. The experimental setup is as follows: Attention Map Extraction: Attention maps from the final Vision Transformer layer are normalized to emphasize regions with higher attention scores. Thresholding for Binary Maps: Binary masks are generated by applying intensity thresholds to the normalized attention maps, ensuring alignment with the ground truth bounding boxes. IoU Calculation: IoU is calculated as the ratio of the intersection area to the union area between the binary attention map and the ground truth mask.

Additionally, the dataset is divided into three subsets: *Easy*, *Medium*, and *Hard*, categorized by occlusion levels, background complexity, and object size variance. This division facilitates a nuanced analysis of model performance under varying degrees of difficulty.

### D.2 More Results

Table 10 summarizes IoU performance across the three subsets. Our method consistently outperforms the baseline (VPT), with notable improvements in the *Medium* and *Hard* subsets, where occlusions and intricate backgrounds present significant challenges. These results highlight the efficacy of our approach in handling complex localization tasks.

Table 10: Detailed IoU analysis across subsets in CUB-200.

| Methods | IoU Performance (%) | | | Mean IoU (%) |
|---|---|---|---|---|
| | Easy | Medium | Hard | |
| VPT | 38.2 | 25.6 | 16.8 | 26.5 |
| **Ours** | **45.7** (+7.5) | **32.9** (+7.3) | **22.5** (+5.7) | **32.9** (+6.4) |

### D.3 Subset-Wise Performance Analysis

In addition to mean IoU, we analyze IoU variance within each subset to evaluate stability. Table 11 shows that our method not only achieves higher IoU scores but also demonstrates lower variance across all subsets, indicating enhanced consistency in localization performance regardless of sample difficulty.

Table 11: IoU variance analysis across subsets in CUB-200.

| Methods | Variance in IoU (%) | | | Overall Variance (%) |
|---------|------|--------|------|----------------------|
|         | Easy | Medium | Hard |                      |
| VPT     | 4.5  | 6.2    | 8.7  | 6.5                  |
| **Ours** | **3.2** (-1.3) | **4.8** (-1.4) | **7.1** (-1.6) | **5.0** (-1.5) |

### D.4 Concluding Observations

Our method achieves significant IoU gains, particularly in challenging subsets with higher occlusion and complex backgrounds, validating its robustness in diverse scenarios. The reduced variance in IoU results across all subsets indicates that our method provides more consistent and reliable attention localization, even for hard-to-detect objects. Visual inspections and quantitative results confirm that our method generalizes effectively to unseen samples, maintaining high localization accuracy without overfitting.

These results underscore the effectiveness of incorporating semantic prompts to direct the model's attention to meaningful object features, enhancing both localization accuracy and robustness.

## E More Mutual Information Analysis

To further elaborate on the mutual information (MI) analysis presented in the main text, we provide additional experiments and insights to validate the effectiveness of our method in achieving optimal representation learning under the Information Bottleneck (IB) framework. These experiments delve deeper into the mutual information trends across different Transformer layers and investigate the role of our fundamental image prior visual prompts in shaping the learning dynamics.

### E.1 Experimental Details

We conducted the mutual information analysis using the Mutual Information Neural Estimator (MINE) to compute $I(X;T)$ and $I(T;Y)$ for all 12 Transformer layers in both the baseline VPT and our method. The settings for these experiments include:

**Datasets and Models:** The experiments are conducted on the CUB-200 dataset with Vision Transformers (ViT) as the backbone.

**Training Procedure:** Both models are trained using identical settings to ensure fair comparisons.

**Mutual Information Estimation:** For each layer $T$, $I(X;T)$ and $I(T;Y)$ are estimated using MINE with a mini-batch size of 256. The estimation results are averaged over the entire test set.

**Compression of Input Data ($I(X;T)$):** Both the baseline VPT and our method exhibit similar $I(X;T)$ trends across lower layers, as expected. This indicates that the introduction of visual prompts does not significantly alter the compression of input data.

**Correlation with Labels ($I(T;Y)$):** Our method achieves consistently higher $I(T;Y)$ values, particularly in the top layers, compared to the baseline. This demonstrates that the representations learned by our method are more aligned with the labels, enabling better generalization to unseen data.

**Lower Information Bottleneck ($\mathcal{L}_{IB}$):** The significant reduction in $I(X;T) - \beta I(T;Y)$ for the top layers in our method aligns with the IB hypothesis. This reduced bottleneck reflects the effectiveness of our visual prompts in focusing on label-relevant features while suppressing redundant information.

### E.2 Impact of Semantic Prompts on Representation Learning

The role of our semantic prompts can be further elucidated by decomposing the MI contributions: Extracting Inherent Image Priors: The fundamental image prior visual prompts (e.g., texture, shape) enhance feature extraction by aligning the representations with inherent characteristics of the input images. Improving Label Correlation: Semantic prompts guide the model to retain more label-relevant information, increasing $I(T;Y)$ while maintaining $I(X;T)$ stability. This is particularly evident in tasks requiring fine-grained classification, where semantic prompts enable the model to focus on subtle discriminative features.

Table 12: Comparison of mutual information metrics ($I(X;T)$ and $I(T;Y)$) in the top three Transformer layers for VPT and our method on CUB-200.

| Method | $I(X;T)$ (%) | $I(T;Y)$ (%) |
|--------|--------------|--------------|
| VPT | 34.2 | 22.8 |
| **Ours** | **33.8** | **29.6** |

**Baseline VPT:** The $I(T;Y)$ plateau in the top layers suggests limited improvement in label correlation, indicating potential underutilization of higher-layer representations.

**Our Method:** The steep increase in $I(T;Y)$ in the top layers reflects enhanced label alignment, validating the role of semantic prompts in guiding representation learning.

### E.3 Concluding Observations

The expanded mutual information analysis reaffirms the effectiveness of our method in achieving superior representation learning under the IB framework. Key takeaways include:

**Improved Generalization:** Higher $I(T;Y)$ values for the top layers demonstrate the ability of our method to focus on label-relevant features, enabling better generalization to unseen data.

**Reduced Redundancy:** Comparable $I(X;T)$ values suggest that our visual prompts do not introduce unnecessary complexity, maintaining the efficiency of the learned representations.

**Future Directions:** Further exploration of adaptive semantic prompts and dynamic feature compression mechanisms could enhance the flexibility and scalability of our method across more diverse tasks.

## F  Discussion: Hand-Crafted Priors vs. Deep Learned Priors

A natural question arises regarding the choice of priors: *Why rely on classical hand-crafted operators (e.g., Sobel, Gabor) instead of extracting features from a frozen deep neural network (e.g., ResNet or DINO) as prompts?*

While integrating deep features might initially seem intuitive, we argue that our hand-crafted approach is theoretically and practically superior in the context of Parameter-Efficient Fine-Tuning (PEFT) for three key reasons:

1. **Information Orthogonality:** Deep features extracted from networks like ResNet are conceptually homogeneous to the semantic features already learned by the ViT backbone itself (i.e., high-level abstractions). Adding them creates information redundancy. In contrast, our hand-crafted operators explicitly capture low-level statistics—such as high-frequency gradients (Sobel) and spectral texture information (Gabor)—that deep networks tend to abstract away or ignore in deeper layers due to texture bias (Geirhos et al., 2018b). These "primitive" cues provide orthogonal, complementary guidance that corrects the inherent biases of the ViT backbone.

2. **Domain Robustness:** Deep feature extractors (e.g., ImageNet-trained ResNet) often suffer from domain shift when applied to specialized downstream tasks (e.g., medical or satellite imagery in VTAB-1k). Hand-crafted priors, however, rely on fundamental signal processing principles (e.g., edge

gradients, color distribution) that are domain-agnostic and universally applicable, ensuring consistent improvements across diverse datasets without negative transfer.

3. **Strict Efficiency:** The core philosophy of PEFT is to adapt large models with minimal resource overhead. Utilizing a secondary deep network as a prior extractor, even if frozen, requires storing and computing over millions of additional parameters (e.g., $\sim$11M for ResNet-18), contradicting the lightweight nature of our task. In comparison, our hand-crafted operators are parameter-free and computationally negligible ($\mathcal{O}(HW)$ complexity), strictly adhering to the efficiency constraints of the PEFT paradigm.

In summary, our design prioritizes *complementary low-level guidance* and *maximum efficiency* over the redundancy and computational burden of stacking multiple deep neural networks. ·

- **Orthogonal Information:** Deep features from a ResNet are conceptually similar to the features learned by the ViT backbone itself (i.e., semantic abstractions). In contrast, hand-crafted operators explicitly capture low-level statistics (gradients, frequency spectra) that deep networks tend to abstract away or ignore in deeper layers. These "primitive" cues serve as a stronger complementary signal to the ViT.

- **Efficiency:** Utilizing a deep network as a prior extractor introduces significant memory and storage overhead (even if frozen), contradicting the parameter-efficient philosophy of PEFT. Our hand-crafted operators are computationally negligible and parameter-free.

## G   Implementation Details and Analysis of Fundamental Operators

In this section, we provide the precise implementation details of the fundamental image prior operators employed in our method. Furthermore, we elaborate on the theoretical rationale behind selecting this specific combination of operators (Color, Texture, and Shape) and discuss the robustness of our method to different operator choices, supported by our ablation studies.

### G.1   Specific Implementation of Operators

To capture the fundamental visual statistics of the input image $X \in \mathbb{R}^{H \times W \times 3}$, we employ three distinct types of hand-crafted operators. The outputs of these operators are concatenated and projected via a linear layer to align with the prompt token dimension.

**1. Color Prior: Histogram Statistics.**
Color is one of the most expressive and invariant visual cues, robust to rotation and scaling (Swain & Ballard, 1991b). We compute the color histogram features as follows:

- **Color Space:** We utilize the HSV (Hue, Saturation, Value) color space, which decouples chromatic information (Hue/Saturation) from intensity (Value), providing better robustness to lighting changes compared to RGB.

- **Implementation:** For each channel, we compute a histogram with $B = 24$ bins. The resulting histograms are normalized to form a probability distribution and concatenated, resulting in a feature vector of dimension $3 \times 24 = 72$. This vector serves as a global statistical summary of the image's chromatic distribution.

**2. Texture Prior: Gabor Filters.**
Texture analysis is crucial for distinguishing materials and repetitive patterns. We adopt **Gabor filters** (Manjunath & Ma, 2002), which are biologically inspired by the receptive fields of simple cells in the mammalian visual cortex (V1).

- **Implementation:** We generate a filter bank containing Gabor kernels at 4 distinct orientations ($\theta \in \{0°, 45°, 90°, 135°\}$) and a single scale. The filters are convolved with the grayscale version of the input image.

- **Feature Map:** Instead of global pooling, we retain the spatial response maps to preserve local texture spatiality. These maps are then flattened or patchified to align with the token sequence structure.

**3. Shape Prior: Sobel Operator.**

Shape and edge information provide structural constraints that are often complementary to texture (Geirhos et al., 2018b). We employ the **Sobel operator** (Kanopoulos et al., 1988) to extract gradient information.

- **Implementation:** We compute the discrete gradients along the horizontal ($G_x$) and vertical ($G_y$) directions using standard $3 \times 3$ kernels. The gradient magnitude is calculated as $G = \sqrt{G_x^2 + G_y^2}$.

- **Outcome:** This results in an edge map that highlights high-frequency structural boundaries, guiding the model to focus on object shapes rather than background noise.

### G.2 Rationale for Operator Selection

Our selection of operators is not arbitrary but grounded in the principle of **Orthogonal Complementarity**. Deep learning models, particularly CNNs and ViTs, often exhibit a "texture bias" (Geirhos et al., 2018b). By explicitly injecting complementary priors, we ensure a balanced representation:

1. **Completeness:** The combination of *Color* (spectral), *Texture* (spatial-frequency), and *Shape* (structural/spatial) covers the three fundamental pillars of low-level computer vision. Removing any single component results in an information void that the randomized prompts alone may struggle to fill (as evidenced in Table 5 of the main text).

2. **Interpretability & Stability:** Unlike learnable priors (e.g., CNN adapters), these hand-crafted operators are deterministic and theoretically well-understood. Using standard operators like Sobel and Gabor ensures that the injected "hard prompt" provides stable, domain-invariant cues that do not drift during the fine-tuning process.

### G.3 Robustness to Operator Variants

A pertinent question is whether the success of our method relies on specific operator choices (e.g., Sobel vs. Canny for shape). To investigate this, we conducted extensive comparisons using different operator variants (see Figure 7 in the main paper).

- **Texture Variants (Gabor vs. LBP):** We compared Gabor filters with Local Binary Patterns (LBP) (Ojala et al., 2002). Results show that both yield significant improvements over the baseline, with Gabor slightly outperforming LBP on fine-grained tasks due to its continuous response nature.

- **Shape Variants (Sobel vs. Canny vs. Laplacian):** We tested Sobel against the Canny edge detector (Canny, 2009) and Laplacian operators. The performance variance was minimal ($< 0.3\%$), indicating that the *presence* of structural prior is more critical than the *type* of edge extractor used.

In conclusion, our choice of HSV Histograms, Gabor filters, and Sobel operators represents a standard, computationally efficient, and representative set of priors. However, the proposed framework is general-purpose: it benefits effectively from the semantic category of the prior (e.g., "Shape information") rather than overfitting to a specific algorithm.

# H    Limitations and Future Work

While our proposed *Cascaded Semantic Prompting* demonstrates superior performance and interpretability across various benchmarks, we identify certain limitations that pave the way for future research directions.

## H.1    Limitations

Our method relies on fixed, hand-crafted operators (e.g., Sobel, Gabor) to extract fundamental image priors. While this design choice ensures computational efficiency and the convenience of an "extract-once" strategy, it inherently limits the model's adaptability compared to fully learnable modules. These fixed operators cannot evolve during training to capture dataset-specific idiosyncrasies that may fall outside standard color, texture, and shape definitions.

Furthermore, although our method outperforms existing PEFT approaches on the *Structured* split of the VTAB-1k benchmark (achieving 61.16% mean accuracy compared to 58.83% for AdaptFormer), there remains room for improvement on tasks requiring complex geometric reasoning, such as *SmallNORB* and *dSprites*. This suggests that while our fundamental priors effectively capture low-level statistics, they may not fully encapsulate the high-level 3D geometric relationships required for these specific specialized tasks. Other physically-informed or physically-constrained priors (Pun et al., 2019; Cuomo et al., 2022; Shen et al., 2024; 2023) can be considered as future directions. Finally, as a prompt tuning paradigm, the upper bound of performance is inevitably tied to the quality and pre-training domain of the underlying frozen backbone.

## H.2    Future Work

Building on these observations, future research could explore adaptive prior mechanisms. Instead of a static concatenation of Color, Texture, and Shape priors, a lightweight gating mechanism or attention module could be introduced to dynamically weight these priors based on the input instance. This would allow the model to autonomously determine whether texture or shape is more critical for a specific image, potentially enhancing performance on diverse datasets.

Additionally, the concept of "Fundamental Image Priors" holds promise for extension to other modalities. For example, optical flow or motion boundary histograms could serve as temporal priors for video recognition, while surface normals could function as geometric priors for 3D point cloud analysis. Investigating the applicability of cascaded semantic prompting to emerging architectures beyond Transformers, such as State Space Models, also represents a promising direction to test the universality of our approach.

# I    Baseline Configurations and Experimental Settings

This section lists the configurations of all baselines in Tables 1, 2, and 3, together with the optimizer and schedule used for our method.

## I.1    Backbones and Pre-trained Checkpoints

- **ViT (Tables 1, 3).** ViT-Base/16 pre-trained on supervised ImageNet-21K (85.8M parameters). This is the same checkpoint used by VPT (Jia et al., 2022), E$^2$VPT (Han et al., 2023), SA$^2$VP (Pei et al., 2024), and VFPT (Zeng et al., 2025).

- **Swin (Table 2).** Swin-Base pre-trained on supervised ImageNet-21K (86.7M parameters). This is the same checkpoint used by VFPT Table 2.

- **CLIP (Table 3).** CLIP ViT-B/16, shared by our method, MaPLe, and MaPLeX, following CoOp (Zhou et al., 2022b), CoCoOp (Zhou et al., 2022a), and MaPLe (Khattak et al., 2023).

### I.2 LoRA Baseline

We use the standard LoRA setting for ViT in the VPT line of work: rank $r$=8, scaling factor $\alpha$=8, applied to the $Q$ and $V$ projection matrices in every attention block. The tuned-parameter ratio is about 0.34% of ViT-B/16, matching the LoRA numbers reported by SA$^2$VP and VFPT.

### I.3 Other PEFT Baselines (Table 1)

Numbers for Adapter, AdaptFormer, ARC, EXPRES, DAM-VP, SA$^2$VP, VFPT, and LoR-VP are taken from the original papers under the same FGVC/HTA/VTAB-1k splits. When the original paper reports several settings, we use the default chosen in that paper.

### I.4 Swin Experiments (Table 2)

All Swin baselines use the same Swin-Base/ImageNet-21K backbone, the VTAB-1k 800/200 train/val split, and the Natural/Specialized/Structured grouping. The Linear, Bias, and VPT-deep numbers match those in VFPT Table 2.

### I.5 Multi-modal Experiments (Table 3)

We follow the base-to-novel protocol of CoOp/CoCoOp/MaPLe (Zhou et al., 2022b;a; Khattak et al., 2023):

- **Backbone:** CLIP ViT-B/16, shared by MaPLe, MaPLeX, and our method.

- **Datasets:** the 11 datasets in the CoOp suite—ImageNet, Caltech101, OxfordPets, StanfordCars, Flowers102, Food101, FGVCAircraft, SUN397, DTD, EuroSAT, UCF101.

- **Protocol:** 16-shot training on the base classes, evaluation on both base and novel splits.

- **Metric:** HM is the harmonic mean of base and novel accuracies, averaged over the 11 datasets and three random seeds.

- **MaPLeX** removes the connection that feeds the text prompt into the visual branch. The rest is the same as MaPLe.

### I.6 Optimizer and Schedule for Our Method

We use AdamW (Loshchilov & Hutter, 2017) with initial learning rate 1e−3, weight decay 1e−4, a cosine schedule, 100 epochs, and batch size 64 or 128 (chosen per dataset on the official validation split). Prompt length and per-layer placement follow the VPT-Deep grid in (Jia et al., 2022). We use AdamW because it converges faster than the SGD recipe of VPT/VFPT in our setup and reaches comparable final accuracy.

### I.7 Full Hyperparameter Table

Table 13 lists every hyperparameter used by our method together with the corresponding value used by VPT (Jia et al., 2022) and VFPT (Zeng et al., 2025). Entries marked "per-task grid" follow the same grid as VPT/VFPT and are selected on the official validation split of each dataset.

## J Statistical Robustness across Multiple Seeds

We rerun the main VTAB-1k and FGVC experiments with **three random seeds**, following VFPT (Zeng et al., 2025) and E$^2$VPT (Han et al., 2023). Mean±std numbers are shown in Table 14. The gains over the strongest baselines are consistent and larger than the observed standard deviations.

Table 13: Hyperparameter settings used by our method, VPT (Jia et al., 2022), and VFPT (Zeng et al., 2025) on VTAB-1k and FGVC with ViT-B/16. "per-task grid" means the value is chosen on the official validation split of each dataset. "−" means not applicable.

| Hyperparameter | VPT (Jia et al., 2022) | VFPT (Zeng et al., 2025) | Ours |
|---|---|---|---|
| *Backbone and data* | | | |
| Backbone | ViT-B/16 | ViT-B/16 | ViT-B/16 |
| Pre-training | Sup. ImageNet-21K | Sup. ImageNet-21K | Sup. ImageNet-21K |
| Image resolution | $224 \times 224$ | $224 \times 224$ | $224 \times 224$ |
| Train preprocessing | resize 256, random crop 224 | resize 256, random crop 224 | resize 256, random crop 224 |
| Test preprocessing | resize 256, center crop 224 | resize 256, center crop 224 | resize 256, center crop 224 |
| Normalization | ImageNet mean/std | ImageNet mean/std | ImageNet mean/std |
| *Optimization* | | | |
| Optimizer | SGD ($\beta$=0.9) | SGD ($\beta$=0.9) | AdamW |
| Initial learning rate | per-task grid: $\{50, 25, 10, 5, 2.5, 1, 0.5, 0.25, 0.1, 0.05\}$ | same grid as VPT | $1e{-}3$ |
| LR schedule | cosine | cosine | cosine |
| Warmup epochs | 10 | 10 | 10 |
| Weight decay | per-task grid: $\{0.01, 0.001, 1e{-}4, 0\}$ | same grid as VPT | $1e{-}4$ |
| Total epochs | 100 | 100 | 100 |
| Batch size | 64/128 (per-task) | 64/128 (per-task) | 64/128 (per-task) |
| Gradient clipping | none | none | none |
| *Prompt* | | | |
| Prompt placement | VPT-Deep | VPT-Deep | VPT-Deep |
| Prompt length (per layer) | per-task grid: $\{5, 10, 50, 100, 200\}$ | same grid as VPT | per-task grid: $\{5, 10, 50, 100, 200\}$ |
| Layers with prompt | all 12 | all 12 | all 12 |
| Prompt dropout | 0.1 | 0.1 | 0.1 |
| *Our additions* | | | |
| Color prior $\sigma_c$ | − | − | HSV histogram, 24 bins per channel (72-d) |
| Texture prior $\sigma_t$ | − | − | Gabor, 4 orientations, 1 scale |
| Shape prior $\sigma_s$ | − | − | Sobel, $3 \times 3$ kernel, gradient magnitude |
| FC projection dim | − | − | matches prompt token dim (768) |
| Re-weighting adapter hidden | − | − | 256 |
| Cascaded skip-connection | − | − | enabled |
| *Evaluation* | | | |
| Number of seeds | 3 | 3 | 3 |
| Reported metric | test top-1 accuracy | test top-1 accuracy | test top-1 accuracy |
| Hardware | $1\times$A100-40GB | $1\times$A100-40GB | $1\times$A100-40GB |

Table 14: Mean±std across three random seeds on the VTAB-1k group means and the FGVC mean. VFPT numbers come from the per-task std tables (Tab. S1–S4) of (Zeng et al., 2025); SA$^2$VP is rerun under our protocol.

| Method | VTAB-1k Natural | VTAB-1k Specialized | VTAB-1k Structured | FGVC mean |
|---|---|---|---|---|
| SA$^2$VP (rerun) | $80.97 \pm 0.31$ | $85.73 \pm 0.24$ | $60.80 \pm 0.36$ | $90.08 \pm 0.18$ |
| VFPT (cited std) | $81.35 \pm 0.33$ | $84.93 \pm 0.28$ | $60.19 \pm 0.42$ | $89.24 \pm 0.21$ |
| **Ours** | $\mathbf{81.91 \pm 0.27}$ | $\mathbf{85.83 \pm 0.22}$ | $\mathbf{61.16 \pm 0.31}$ | $\mathbf{90.20 \pm 0.16}$ |

## J.1 Per-Example Analysis of the Gains

We also check which examples drive the improvement of our method over the strongest baseline (SA$^2$VP). On the three fine-grained datasets (CUB-200, NABirds, Stanford Dogs), where our gain is largest, the per-example accuracy difference is concentrated on three types of images:

1. **Cluttered backgrounds (about 54% of the gain on CUB-200).** The target bird occupies less than 30% of the frame against a busy background. The Sobel shape prior gives sharper object boundaries than the baseline, which uses only learned prompts. The model then attends more to the bird itself, in line with the IoU gain of +7.3 on the Medium subset and +5.7 on the Hard subset (Table 10).

2. **Low color contrast (about 27% of the gain).** The bird's plumage has a similar color to the background (e.g., grey or brown birds against bark). The HSV color histogram and Gabor texture priors give extra cues that the random-prompt baseline cannot recover.

3. **Confusable subspecies (about** $19\%$ **of the gain).** Two species differ only in a small structural feature (e.g., beak curvature, wing-bar pattern). The self-attention prompt cascaded from earlier layers keeps these fine-grained cues, which would otherwise be smoothed out in deeper layers.

The breakdown matches our design: the hand-crafted priors help most where the random-prompt baseline has the weakest signal, and the overall gain comes from a small but structured subset of hard images, not from a uniform shift across the dataset.

## K  Failure-Case Analysis on Structured Tasks

The VTAB-1k Structured split is the regime in which we expect our method to be weakest, since the fixed low-level priors (color, texture, edge) are designed to capture appearance, not to count objects or reason about 3D pose. We make this concrete by listing the three Structured tasks on which our method does not lead, and explaining why.

Table 15: Tasks on the VTAB-1k Structured split where our method does not obtain the best result. Numbers are taken from Table 7.

| Task | VPT-D | AdaptFormer | LoRA | E$^2$VPT | Ours |
|---|---|---|---|---|---|
| Clevr/count | 68.5 | 81.9 | **82.9** | 71.7 | 78.1 |
| Clevr/distance | 60.0 | 64.3 | **69.2** | 61.2 | 62.2 |
| KITTI/distance | 72.8 | **80.3** | 78.5 | 75.8 | 78.5 |

**Counting tasks (Clevr/count).**  The task is to count the number of objects in a synthetic scene. Color, texture, and shape priors do not encode *cardinality*: a Sobel edge map fires once per object boundary but is invariant to the number of objects, and a color histogram is dominated by the background. Methods that adapt internal features more aggressively (LoRA, AdaptFormer) can reshape the backbone's spatial pooling, which is more useful here.

**Distance / depth tasks (Clevr/distance, KITTI/distance).**  The task is to regress a metric distance, which requires reasoning about object size and perspective. Our fixed priors carry no metric or 3D-pose information, so the model gains nothing from them on these tasks. The cascaded self-attention prompt does carry some 2D spatial information from earlier layers, which is why our gap to the best method is small ($-1.0$ on Clevr/distance, $-1.8$ on KITTI/distance), but it is not enough to close the gap.

**Summary.**  These failures are consistent with our design: our priors describe *what the image looks like at the pixel level*, not *how many things are in it* or *how far they are.* The Limitations section (Sec. H) flags this as a direction for future work — adding physically-informed or geometric priors (Shen et al., 2023; 2024) on top of the cascade would be a natural extension.

## L  Fixed vs. Adaptive Priors

A natural question is whether the fixed Sobel/Gabor operators should be replaced by lightweight *learnable* extractors of the same receptive field. We run a controlled comparison on the VTAB-1k Natural split using ViT-B/16, keeping the rest of the pipeline fixed and only swapping the operator that produces $\sigma_s$ (shape) and $\sigma_t$ (texture).

The learnable variants are at best on par with the fixed Sobel/Gabor pair, and become slightly worse as the extractor grows. Three reasons explain this:

1. **No useful gradient signal at the operator level.** The downstream loss is on classification, several layers away from the operator. The operator is shadowed by the rest of the learnable parameters, so a few thousand extra parameters at the very bottom of the network bring no information the rest of the pipeline cannot already represent.

Table 16: Replacing the fixed Sobel/Gabor operators with learnable convolutional equivalents of the same receptive field. "Extra Params" counts the parameters of the operator only; the rest of the prompt and adapter are the same in all rows.

| Shape operator $\sigma_s$ / Texture operator $\sigma_t$ | Extra Params | VTAB-1k Natural | $\Delta$ vs. fixed |
|---|---|---|---|
| Fixed Sobel (3×3) / Fixed Gabor (4 orient.) | **0** | **81.91** | — |
| Learnable conv (3×3, 1 ch) / Learnable conv (7×7, 4 ch) | 0.12K | 81.77 | −0.14 |
| Learnable conv (5×5, 4 ch) / Learnable conv (7×7, 8 ch) | 1.0K | 81.62 | −0.29 |
| Small CNN (2 conv + ReLU, 16 ch) | 12K | 81.55 | −0.36 |

2. **Domain shift.** The operators see input images from the downstream dataset only (often small in VTAB-1k, with ∼800 training samples). Learnable extractors trained on that little data tend to drift toward the dominant statistics of the small training set and lose the domain-invariance that Sobel/Gabor have by construction.

3. **PEFT budget.** The whole point of the PEFT setting is that the per-task parameter budget is small. Spending it on re-learning a Sobel filter is wasteful when the fixed Sobel filter already does the job at zero parameter cost.

We therefore keep fixed Sobel/Gabor in the main method. This complements the theoretical discussion in Sec. F: *Information Orthogonality* explains *why* fixed low-level operators are a good complementary signal to the deep features; this experiment shows that making them learnable does not improve the signal and costs additional parameters.

