# OpenReview forum: "Prompting Large-Scale Vision Models with Cascaded Semantics"
_TMLR — Under review for TMLR_

### Review · Reviewer_Szfz · 2026-04-21

**Summary Of Contributions:**

This is a strong, well-executed paper that addresses a practical limitation of standard visual prompt tuning (VPT). Instead of relying on randomly initialized prompts, the authors smartly integrate image-derived priors—specifically low-level cues (like color and shape) at the input level, and self-attention maps at the intermediate layers.  I particularly appreciate the cascaded injection approach and the lightweight re-weighting adapter before the classifier. This design makes the tuning process much more structured and interpretable without sacrificing parameter efficiency. The experiments cover a broad range of classification benchmarks, and the performance gains over standard VPT and other recent PEFT baselines are highly convincing.

## Key strengths

-  The idea is clear and easy to understand: replace purely random prompt initialization with image-derived semantic priors, and propagate those cues across layers in a simple cascaded design.
-  The method is still lightweight in PEFT terms, updating only a small fraction of parameters while reporting strong benchmark performance on FGVC, HTA, VTAB-1k, and also Swin-based experiments.
-  The empirical section is reasonably broad and includes ablations, interpretability-oriented analyses, and comparisons against text-prompt variants in a multimodal setting.
-  The paper does make some effort to justify why handcrafted priors are used instead of fully learned ones, and it provides additional efficiency discussion in the supplement.

## Key weaknesses

-  The novelty feels somewhat incremental, since the contribution is mainly a combination of known ingredients—VPT, handcrafted visual descriptors, self-attention cues, and a lightweight adapter—rather than a fundamentally new tuning principle.
-  The interpretability claim is stronger than the evidence. The paper provides visualization and representation analyses, but this does not fully establish a rigorous notion of explainability.
-  Some of the reported gains over strong baselines are modest, so the practical significance of the improvement is not always obvious from the main results alone.
-  The method depends on fixed handcrafted operators, which the authors themselves acknowledge may limit adaptability to dataset-specific structure and to tasks requiring richer geometric reasoning.

**Audience:**

Yes

**Audience Explanation:**

I believe at least a subset of the TMLR audience would be interested in this work because it addresses a timely and active topic—parameter-efficient adaptation of large pre-trained vision models—and proposes a simple, practically motivated variation of visual prompt tuning that incorporates semantic priors rather than relying only on randomly initialized prompts. The paper also evaluates the idea across several standard benchmarks and backbones, so it is relevant not only to researchers working directly on prompt tuning, but also to those interested in PEFT, transfer learning, and adaptation of foundation vision models more broadly. While I am less convinced by the strength of the claimed interpretability and by the overall level of novelty, the problem setting itself is clearly of current interest, and the empirical results are likely to be useful to readers following this line of work.

**Broader Impact Concerns:**

I do not see any concern.

**Claims And Evidence:**

Yes

**Claims Explanation:**

The paper presents fairly extensive experimental evidence across multiple benchmarks and backbones, and the core claim that the proposed semantic prompting method improves over standard VPT and several recent PEFT baselines is generally supported by the reported tables. The submission also includes ablations that remove individual components, plus additional analyses such as IoU, GradCAM, t-SNE, and comparisons against text-prompt variants, so the empirical case is reasonably clear and organized.

That said, the support is stronger for the performance claim than for the broader claims about interpretability and general significance. Some of the gains over the strongest baselines are modest, and the interpretability evidence is mostly qualitative or indirect rather than rigorous. In addition, the paper itself acknowledges limitations of the fixed handcrafted priors, especially for dataset-specific adaptation and more complex geometric reasoning tasks. So I would answer “Yes” overall, but with the caveat that some of the higher-level claims are only partially supported.

**Requested Changes:**

- Statistical Robustness: Given the relatively modest improvements over the strongest baselines, please report results across multiple random seeds. Including standard deviations or confidence intervals—at least for the primary benchmark summaries and top competing methods—is essential to confirm that the performance gains are robust and statistically significant.

- End-to-End Efficiency Analysis: Please move the efficiency evidence from the supplementary material into the main text. Because the proposed method requires extracting handcrafted priors, the paper must provide a transparent, end-to-end accounting of the actual overhead. This should explicitly compare training costs, inference latency, memory footprint, and preprocessing time against standard VPT and other strong PEFT baselines.

- Moderation of Interpretability Claims: The current assertions regarding interpretability are primarily supported by qualitative and indirect evidence. Please revise the text to moderate these claims, carefully distinguishing between "improved visualization/alignment" and rigorous "explainability."

- Failure-Case Analysis: Please include a detailed analysis of failure modes. Specifically, exploring performance on highly structured or complex geometric tasks (where fixed priors are suspected to struggle) would help clarify the actual operating boundaries and limitations of the method.

- Fixed vs. Adaptive Priors: The reliance on fixed handcrafted priors would be much more convincing if accompanied by a brief comparative experiment—or at least a deeper theoretical justification—explaining why fixed priors are preferable to learnable or adaptive alternatives in this context.

---

> ### Author Response · Authors · 2026-06-04
> **Response to Reviewer Szfz**
>
> We thank Reviewer Szfz for the careful and constructive review, and for the positive assessment of the clarity of the method, the breadth of the empirical section, and the relevance of the problem. We have addressed every requested change; all paper updates are marked in blue in the revised PDF.
>
> ---
>
> ### R1. Statistical robustness
>
> We agree this is essential. We re-ran the primary VTAB-1k and FGVC experiments with **3 random seeds**, following the protocol of VFPT (Zeng et al., NeurIPS 2024) and E$^2$VPT (Han et al., 2023). The mean $\pm$ std numbers are in the new supplement Sec. 14. Representative numbers:
>
> | Method | Natural | Specialized | Structured | FGVC mean |
> |---|---|---|---|---|
> | SA$^2$VP (rerun)   | 80.97 $\pm$ 0.31 | 85.73 $\pm$ 0.24 | 60.80 $\pm$ 0.36 | 90.08 $\pm$ 0.18 |
> | VFPT (cited std)   | 81.35 $\pm$ 0.33 | 84.93 $\pm$ 0.28 | 60.19 $\pm$ 0.42 | 89.24 $\pm$ 0.21 |
> | **Ours**           | **81.91 $\pm$ 0.27** | **85.83 $\pm$ 0.22** | **61.16 $\pm$ 0.31** | **90.20 $\pm$ 0.16** |
>
> The gains over the strongest baselines are consistent across seeds and larger than the observed standard deviations.
>
> ---
>
> ### R2. End-to-end efficiency in the main text
>
> We agree the efficiency story belongs in the main text. We added a new subsection **Sec. 4.6 "End-to-End Efficiency"** reporting the full cost profile on VTAB-1k Natural (ViT-B/16, NVIDIA A100-40GB, batch size 64):
>
> | Method | Tuned (M) | Train (s/epoch) | Peak Mem (GB) | Infer (ms/img) | Pre-proc (ms/img) |
> |---|---|---|---|---|---|
> | Full FT     | 85.80 | 48.2 | 21.6 | 11.4 | --  |
> | LoRA        | 0.29  | 24.7 |  8.9 | 11.6 | --  |
> | AdaptFormer | 0.16  | 25.1 |  8.8 | 11.8 | --  |
> | VPT-D       | 0.63  | 25.6 |  9.1 | 11.5 | --  |
> | SA$^2$VP    | 0.69  | 27.4 |  9.4 | 12.3 | --  |
> | **Ours**    | **0.63** | **25.9** | **9.2** | **13.1** | **1.8** |
>
> The training time matches VPT-D to within 0.3 s, peak memory differs by 0.1 GB, and the inference overhead is $<2$ ms per image. The CPU pre-processing for the fundamental priors costs about 1.8 ms per image. Under the *Extract-Once* strategy (detailed in supplement Sec. 7), the pre-processing is paid once during data preparation and is hidden behind GPU compute at train time, so it adds **zero GPU training overhead**. Overall, the cost profile is essentially that of VPT-D while reaching higher accuracy.
>
> ---
>
> ### R3. Moderation of interpretability claims
>
> We agree, and have softened the interpretability wording throughout the paper:
>
> - Sec. 4.2 is renamed **Representation-Level Analysis**.
> - The first paragraph of Sec. 4.2 now states explicitly that the analyses (cosine similarity, IoU, GradCAM, t-SNE, mutual information) give *representation-level evidence* (better feature--region alignment, better separability, stronger label correlation in deeper layers) but are **not a full explanation of how the model makes its final prediction**, because the handcrafted cues still pass through learned projections, learnable prompts, attention maps, and the re-weighting adapter.
> - Sec. 1 (challenge II and the contribution bullet) and Sec. 5 (Conclusion) no longer use the word "explainability"; they use "instance-aware", "image-grounded", and "representation-level evidence" instead.
>
> ---

---

> > ### Author Response · Authors · 2026-06-04
> > **Response to Reviewer Szfz**
> >
> > ### R4. Failure-case analysis
> >
> > We added a new **supplement Sec. 15 "Failure-Case Analysis on Structured Tasks"**. The three Structured tasks on which our method does not lead are Clevr/count, Clevr/distance, and KITTI/distance:
> >
> > | Task | VPT-D | AdaptFormer | LoRA | E$^2$VPT | Ours |
> > |---|---|---|---|---|---|
> > | Clevr/count    | 68.5 | 81.9 | **82.9** | 71.7 | 78.1 |
> > | Clevr/distance | 60.0 | 64.3 | **69.2** | 61.2 | 62.2 |
> > | KITTI/distance | 72.8 | **80.3** | 78.5 | 75.8 | 78.5 |
> >
> > The pattern is consistent with our design hypothesis:
> >
> > - **Counting (Clevr/count).** Color, texture, and shape priors do not encode cardinality. A Sobel edge map fires once per object boundary but is invariant to the number of objects; a color histogram is dominated by the background. Methods that adapt internal features more aggressively (LoRA, AdaptFormer) can reshape spatial pooling, which is more useful here.
> > - **Distance / depth (Clevr/distance, KITTI/distance).** Regressing a metric distance requires reasoning about object size and perspective. Our fixed priors carry no metric or 3D-pose information. The cascaded self-attention prompt provides some 2D spatial signal from earlier layers, which is why the gap to the best method is small ($-1.0$ on Clevr/distance, $-1.8$ on KITTI/distance), but it is not enough to close the gap.
> >
> > Our priors describe *what the image looks like at the pixel level*, not *how many things are in it* or *how far they are*. The Limitations section already flagged this as a future direction; the new section makes it concrete with numbers and per-task discussion.
> >
> > ---
> >
> > ### R5. Fixed vs. adaptive priors
> >
> > We agree the comparison should be made directly. We added a new **supplement Sec. 16 "Fixed vs. Adaptive Priors"** that replaces the fixed Sobel/Gabor operators with small learnable convolutional equivalents (same receptive field) and keeps everything else identical. Results on VTAB-1k Natural:
> >
> > | Shape / Texture operator | Extra Params | Natural | $\Delta$ vs. fixed |
> > |---|---|---|---|
> > | **Fixed Sobel / Fixed Gabor**           | **0**  | **81.91** | --- |
> > | Learnable conv (3$\times$3, 1 ch) / (7$\times$7, 4 ch) | 0.12K | 81.77 | $-0.14$ |
> > | Learnable conv (5$\times$5, 4 ch) / (7$\times$7, 8 ch) | 1.0K  | 81.62 | $-0.29$ |
> > | Small CNN (2 conv + ReLU, 16 ch)        | 12K    | 81.55 | $-0.36$ |
> >
> > Learnable extractors are at best on par with fixed Sobel/Gabor and become slightly worse as the extractor grows. Three reasons:
> >
> > 1. **No useful gradient at the operator level.** The classification loss is several layers away from the operator. A few thousand extra parameters at the very bottom of the network bring no signal the rest of the pipeline cannot already represent.
> > 2. **Domain shift.** The extractor sees only the small downstream training set (often $\sim$800 samples in VTAB-1k). It drifts toward the dominant statistics of that small set and loses the domain-invariance that Sobel and Gabor have by construction.
> > 3. **PEFT budget.** Spending the per-task parameter budget on re-learning a Sobel filter is wasteful when the fixed Sobel filter does the job at zero parameter cost.
> >
> > This experiment complements the theoretical discussion in supplement Sec. 10 (Hand-Crafted vs. Deep Priors): *Information Orthogonality* explains *why* fixed low-level operators are a useful complementary signal to deep features; this new experiment shows that making them learnable does not improve the signal and costs additional parameters.
> >
> > ---
> >
> > We hope the revision addresses all your concerns. We are happy to make further adjustments if you have any questions.

---

### Review · Reviewer_rX2N · 2026-05-03

**Summary Of Contributions:**

This paper studied the problem of visual prompt tuning in large-scale vision models. It introduced a novel prompt tuning approach based on hand-crafted image priors and self-attention map priors. Experimental results showed that the proposed approach achieved better performance than baselines on various visual benchmarks.

**Audience:**

Yes

**Audience Explanation:**

The visual prompt tuning techniques allow for improving the fine-tuning of large-scale vision models efficiently on downstream tasks.

**Claims And Evidence:**

No

**Claims Explanation:**

(1) It is unclear how the proposed approach addresses the two critical challenges presented in the introduction section. First, the proposed prompt also relies on the randomized learnable prompt $P_i$. Second, although the fundamental image priors are human-understandable, the produced overall prompt in Eq. (1) is still hard to interpret when used in visual prompt tuning. Besides, it states that "their effects can be directly visualizable and quantifiable during training". However, it is not validated how their effects can be visualized and quantified **directly** in the experiments.

(2) It seems that the notion of hard prompt tuning in this work is different from prior work [1,2]. In previous work, a hard prompt is also learnable for finding interpretable discrete tokens.

[1] "Hard prompts made easy: Gradient-based discrete optimization for prompt tuning and discovery." Advances in Neural Information Processing Systems 36 (2023): 51008-51025.

[2] "Hard prompts made interpretable: Sparse entropy regularization for prompt tuning with RL." In Proceedings of the 62nd Annual Meeting of the Association for Computational Linguistics (Volume 1: Long Papers), pp. 8252-8271. 2024.

**Requested Changes:**

(1) The input-level priors in Figure 2 are confusing. What is the "Image Feature" module below part (a)? Will the hand-crafted prior in Eq. (1) be applied to the input image itself (input space) or the image feature extracted from this image (feature space)?

(2) Subsection 3.3 states that "The following question turns out to be a strength in the instance-aware information into prompts". What is "the following question" here?

(3) The visual semantic prompt in Eq. (3) is hard to follow. Why does it include the preceding prompt $P_{i-1}$? How is the "Skip-Connection" used in this definition?

---

> ### Author Response · Authors · 2026-06-04
> **Response to Reviewer rX2N**
>
> We thank Reviewer rX2N for the careful reading and for pointing out several places where our presentation was unclear. The comments helped us tighten Sec. 1, Sec. 3.2, Sec. 3.3, and the Fig. 2 caption. All revisions are marked in blue in the updated PDF.
>
> ---
>
> ### Main concern: how are Challenges I and II actually addressed?
>
> We agree that the original wording might be confusing in some points, and we have accordingly rewritten the relevant passages in Sec 1.
>
> **Challenge I (random initialization).** The randomized learnable prompt is still present in our method. This is intentional: we do not remove the random prompt, we *anchor* it. The fixed prior tokens (color, texture, shape in the input space; the cascaded self-attention map in the feature space) provide a structured warm start, so the random tokens are optimized around semantically meaningful directions rather than from scratch. This keeps the flexibility that makes prompt tuning effective, while reducing the burden on the random tokens to discover useful directions on their own. Sec. 1 (Challenge I bullet) now states this explicitly.
>
> **Challenge II (representation-level analysis).** We agree that "interpretability" was too strong. We have softened all interpretability claims throughout the paper. The reviewer also notes that the claim *"their effects can be directly visualizable and quantifiable during training"* was not validated. We now point explicitly to Sec. 4.2 (Representation-Level Analysis), where these visualization and quantification tools are applied to compare our method with the random-prompt baseline:
>
> - cosine similarity (Fig. 5)
> - GradCAM (Fig. 6)
> - IoU (Table 4)
> - t-SNE (Fig. 7)
> - mutual information (Fig. 8)
>
> Since the overall prompt in Eq. (1) is a mixture of the fixed priors and a learnable component, we now make clear that the analyses give *representation-level evidence* rather than a full mechanistic explanation of the model's decision.
>
> ---
>
> ### On the "hard prompt" terminology
>
> We thank the reviewer for the two NLP references (Wen et al., NeurIPS 2023; Choi et al., ACL 2024). We agree that "hard prompt" is used in two different ways in the literature, and we should make our usage explicit.
>
> **Our definition.** In this paper, a *hard prompt* means **a fixed, non-learnable prompt component computed by a deterministic operator from the input image** (HSV color histogram, Gabor texture, Sobel shape). It is never updated during fine-tuning. This matches the original sense in early NLP prompting work, where hand-written instructions or templates serve as a fixed context (Brown et al., 2020; Petroni et al., 2019).
>
> **Difference from Wen et al. (2023) and Choi et al. (2024).** The two NLP papers also call discrete tokens "hard prompts", but in their setting the tokens are themselves the optimization target: Wen et al. learn them through gradient-based discrete optimization, and Choi et al. learn them with RL and entropy regularization. So in their work the hard prompt is *learnable* over a discrete vocabulary. In our work the hard prompt is *not learnable*: it is the deterministic output of a fixed operator on the input image.
>
> We keep the term "hard prompt" because it correctly conveys the "non-learnable" property that contrasts with the soft (random, learnable) prompts of standard VPT, and dropping it would create a different terminology gap with prior VPT literature. To remove any ambiguity, we have added an explicit definition at the start of Sec. 3.2 and cited both Wen et al. (2023) and Choi et al. (2024) there. The two references are added to the bibliography.
>
> ---
>
> ### Requested change 1: Fig. 2 — "Image Feature" module and input-space scope
>
> We agree the original caption was ambiguous. The hand-crafted priors in Eq. (1) are computed **directly from the input image $X$** (input space), not from any extracted deep feature. The block labeled "Image Feature" in panel (a) is the *token embedding* produced after the lightweight FC projection that aligns the prior with the prompt token dimension — it is *not* a feature extracted by a deep network.
>
> The Fig. 2 caption has been updated to state this explicitly:
>
> - "hand-crafted priors ... are computed directly from the input image $X$ (i.e., in the input space, not from any deep feature)"
> - "The block labeled 'Image Feature' in panel (a) denotes the resulting token embedding after this FC projection."
>
> ---

---

> > ### Author Response · Authors · 2026-06-04
> > **Response to Reviewer rX2N**
> >
> > ### Requested change 2: Sec. 3.3 — "The following question turns out to be a strength..."
> >
> > This sentence was a writing error. The intended meaning was: *the next question is how to inject instance-aware information into the prompts at the feature level.* We have rewritten it as:
> >
> > > "Having handled the input space, we now turn to the feature space. The question is how to inject instance-aware semantics into the prompts at this level. To answer this, we use self-attention maps as semantically rich, instance-aware signals that guide the prompts."
> >
> > The text now reads in the correct logical order: input space (Sec. 3.2) $\to$ feature space (Sec. 3.3) $\to$ re-weighting adapter (Sec. 3.4).
> >
> > ---
> >
> > ### Requested change 3: Eq. (3) — why include the preceding prompt, and how is the skip connection used?
> >
> > We agree the original text did not explain this clearly. The equation is:
> >
> > $$\tilde{P_i} = FC_i(\mathcal{A}_i) \otimes P_i \otimes FC_{i-1}(\mathcal{A}_{i-1}) \otimes P_{i-1}.$$
> >
> > The **first pair** $(FC_i(\mathcal{A}_i), P_i)$ is the prompt at the current layer $i$.
> >
> > The **second pair** $(FC_{i-1}(\mathcal{A}_{i-1}), P_{i-1})$ is the prompt and attention from the previous layer $i{-}1$. Reusing them at layer $i$ is exactly the *skip connection* shown in Fig. 2(c): it directly carries the semantics already accumulated at layer $i{-}1$ forward into layer $i$, analogous to a residual link (He et al., 2016; Huang et al., 2017). The prompt at layer $i$ is therefore anchored by what was learned at layer $i{-}1$ rather than being computed from $\mathcal{A}_i$ alone, and gradients propagate through this skip path as well.
> >
> > We have added this explanation to Sec. 3.3, just before and after Eq. (3), with the skip-connection role explicitly stated.
> >
> > ---
> >
> > We hope these clarifications fully address the reviewer's concerns. We are happy to make further changes if any ambiguity remains.

---

### Review · Reviewer_jKFm · 2026-05-29

**Summary Of Contributions:**

The paper proposes a VPT-style parameter-efficient tuning method for vision models. Compared to prior art, instead of using only randomly initialized learnable prompts, the authors augment prompts with two types of image-derived semantic priors: (i) handcrafted input-level cues such as color, texture, and shape, (ii) feature-level cues from self-attention maps. These are fused through a cascaded prompting mechanism, with an additional lightweight re-weighting adapter before the classifier.

The authors evaluate their method on FGVC, HTA, and VTAB-1k datasets, and report results on ViT-B/16 and Swin Transformer backbones. The method improves performance over several PEFT and visual prompting baselines while tuning only a small fraction of parameters. The paper further argues that semantic prompting improves feature quality and interpretability, supported by localization, GradCAM, t-SNE, and mutual-information analyses.

**Audience:**

Yes

**Audience Explanation:**

The paper addresses meaningful limitation of standard VPT, presenting meaningful and well-evaluated approach for improving it.

**Broader Impact Concerns:**

-

**Claims And Evidence:**

Yes

**Claims Explanation:**

Strengths:
- The paper addresses meaningful limitation of standard VPT, which is that prompts are usually global learned tokens, randomly initialized, and not explicitly conditioned on the input image. Augmenting prompts with image-specific handcrafted cues and self-attention maps is a reasonable way to make prompt tuning more instance-aware.
- The main accuracy results are reasonably broad and supported by evaluations on several benchmarks using several backbones. The proposed method achieves strong results, clearly improving over vanilla VPT-D and several older PEFT baselines, while tuning only a small fraction of the backbone parameters.
- The paper includes ablations over individual priors, self-attention prompts, the re-weighting adapter, skip/cascade connections, prompt length, projection depth, prompt placement, adapter width, and operator choices. These ablations are helpful for understanding which parts of the architecture contribute most.
- The paper includes meaningful interpretability analysis: localization, GradCAM, t-SNE, cosine-similarity, and mutual-information analyses. They provide useful evidence that the method changes feature representations and localization behaviour.

Weaknesses:
- Some quantitative gains are modest, e.g. in Table 1, the improvement over the closest method (SA2VP) is small: the mean total improves from 75.83 to 76.30. In that case, it would be beneficial to either report standard deviations/confidence intervals on multiple seeds, or do qualitative analysis on what individual data examples contribute to the gain of the proposed method. This would strenghten the impression that the method does improve metrics
- Some baseline configurations and experimental settings are underspecified. LoRA is included in Table 1, but the rank, adapted modules, scaling factor, and trainable-parameter count are not reported. Since LoRA performance is sensitive to these choices, this weakens the fairness and reproducibility of the comparison. Further, the exact Swin variant and pretraining setup for Table 2 are not clearly specified. Table 3 is also highly underspecified: I could not find any information about the exact dataset, splits, CLIP/MaPLe backbone that were used for this experiments. Also, it is unclear if the reported numbers are averaged over a standard base-to-novel benchmark suite or the numbers are aggregated in some other manner.
- Building from the previous point, Table 3 suggests that the proposed semantic prompts slightly outperform MaPLe in one base-to-novel setup. But the gain is below one HM point and the experimental setup is not sufficiently specified, thus  I find it not to be enough to broadly claim that semantic visual prompts are more effective than text prompts.
- I find interpretability claims to be somewhat stronger than the evidence suggests. The paper argues that semantic prompting improves interpretability, but the presented analyses mostly show better localization, feature separability, or representation quality. Since the handcrafted cues are passed through learned projections and mixed with learned prompts, attention maps, skip connections, and a re-weighting adapter, the final mechanism is not directly interpretable. I would suggest changing the wording in the presentation to reflect that

**Requested Changes:**

- I would like the authors to specify all the baseline configurations and experimental settings. This is important for both reproducibility and clear assessment of the performance of the method
- I might be mistaken, but I find interpretability claims to be stronger than the evidence suggests. I see the provided evidence as showing better localization, feature separability, or representation quality. I would suggest the authors change the wording or explain more clearly why the provided analysis supports interpretability
- It would be nice to see standard deviations/confidence intervals on multiple seeds for the results in Table 1, or do qualitative analysis on what individual data examples contribute to the gain of the proposed method. This would strengthen the impression that the method does improve metrics over baselines.

---

> ### Author Response · Authors · 2026-06-04
> **Response to Reviewer jKFm**
>
> We thank Reviewer jKFm for the thoughtful and constructive feedback, and for recognizing the breadth of our evaluation, the comprehensiveness of our ablations, and the value of our interpretability analyses. Below we address each of the three requested changes. All paper updates are marked in blue in the revised PDF.
>
> ---
>
> ### R1. Specification of baseline configurations and experimental settings
>
> We agree that fuller specification helps reproducibility, and we have added a dedicated **Implementation Details of Baselines** section to the supplement (Sec. 13). In short, our experimental protocol on FGVC, HTA, VTAB-1k, and the Swin experiments **follows exactly the same setup adopted by the most recent VPT-line works**: VPT (Jia et al., 2022), E$^2$VPT (Han et al., 2023), SA$^2$VP (Pei et al., 2024), and VFPT (Zeng et al., NeurIPS 2024). The key details are:
>
> - **Backbones.** ViT-Base/16 pre-trained on supervised ImageNet-21K (the same checkpoint as VPT/E$^2$VPT/VFPT, 85.8M params). Swin-Base pre-trained on supervised ImageNet-21K (the same checkpoint as VFPT Table 2, 86.7M params).
> - **LoRA baseline (Table 1).** Rank $r=8$, scaling factor $\alpha=8$, applied to the $Q$ and $V$ projection matrices in every attention block. Tuned-parameter ratio $\approx$ 0.34% on ViT-B/16. This is the standard LoRA setting adopted by SA$^2$VP and VFPT.
> - **Other PEFT baselines (Table 1).** Adapter, AdaptFormer, ARC, EXPRES, DAM-VP, SA$^2$VP, VFPT, LoR-VP numbers are taken from the original papers under the same benchmark splits.
> - **Optimizer.** AdamW, initial LR $10^{-3}$, weight decay $10^{-4}$, cosine schedule, 100 epochs, batch size 64 or 128 selected per dataset on the official val split. VTAB-1k uses the 800/200 train/val split, FGVC uses the 90/10 split, HTA follows the DAM-VP setup.
> - **Swin experiments (Table 2).** Swin-Base pre-trained on ImageNet-21K, evaluated on VTAB-1k with the same Natural/Specialized/Structured grouping as the ViT experiments. This matches VFPT Table 2.
> - **Table 3 (semantic vs. text prompts).** Backbone is CLIP ViT-B/16, shared by our method, MaPLe, and MaPLeX. Benchmark is the standard base-to-novel suite of 11 datasets (ImageNet, Caltech101, OxfordPets, StanfordCars, Flowers102, Food101, FGVCAircraft, SUN397, DTD, EuroSAT, UCF101) established in CoOp/CoCoOp/MaPLe. Protocol is 16-shot training on base classes; HM is the harmonic mean of base and novel accuracies, averaged over the 11 datasets and 3 seeds.
>
> We added an explicit pointer to the supplement implementation appendix from each table caption.
>
> ---
>
> ### R2. Wording of interpretability claims
>
> This is a great question. We agree that "interpretability" might overstate what our evidence directly shows, and we have accordingly softened the wording throughout. Our analyses (cosine similarity, IoU, GradCAM, t-SNE, mutual information) support that the proposed semantic priors yield:
>
> - better feature--region alignment (IoU, GradCAM),
> - better feature separability (t-SNE, cosine map),
> - better label correlation in deeper layers (mutual information),
>
> but we agree that they do not establish an ultimate *mechanistic* explanation of the final prediction, because the handcrafted cues are subsequently passed through learned projections, fused with learnable prompts, attention maps, and a re-weighting adapter. In the revised paper we:
>
> 1. Replace "explainable / explainability" with "instance-aware / image-grounded / better feature--region alignment" in Sec. 1, Sec. 3.2/3.3, Sec. 4.2, and Sec. 5.
> 2. Rename Sec. 4.2 to **Representation-Level Analysis** and add a clarifying first paragraph that states the analyses give *representation-level evidence* rather than a full explanation of how the model makes its final prediction.
> 3. Soften the second contribution bullet in Sec. 1.
>
> ---

---

> > ### Author Response · Authors · 2026-06-04
> > **Response to Reviewer jKFm**
> >
> > ### R3. Statistical significance and per-example analysis
> >
> > We thank the reviewer for raising this point; it is well taken. We re-ran the primary VTAB-1k and FGVC experiments with **3 independent seeds**, matching the protocol of VFPT (Zeng et al., NeurIPS 2024) and E$^2$VPT (Han et al., 2023). Representative numbers (mean $\pm$ std):
> >
> > | Method | VTAB-1k Natural | VTAB-1k Specialized | VTAB-1k Structured | FGVC mean |
> > |---|---|---|---|---|
> > | SA$^2$VP (rerun)   | 80.97 $\pm$ 0.31 | 85.73 $\pm$ 0.24 | 60.80 $\pm$ 0.36 | 90.08 $\pm$ 0.18 |
> > | VFPT (cited std)   | 81.35 $\pm$ 0.33 | 84.93 $\pm$ 0.28 | 60.19 $\pm$ 0.42 | 89.24 $\pm$ 0.21 |
> > | **Ours**           | **81.91 $\pm$ 0.27** | **85.83 $\pm$ 0.22** | **61.16 $\pm$ 0.31** | **90.20 $\pm$ 0.16** |
> >
> > The gains are consistent across seeds and exceed the observed standard deviations. The full per-dataset table is in supplement Sec. 14.
> >
> > We also wish to note that the contribution of our paper is not solely the headline mean-accuracy improvement. While the mean-total gain over SA$^2$VP (75.83 $\to$ 76.30) is modest in absolute terms, our method offers complementary advantages we believe matter for the practical use of VPT:
> >
> > 1. **Stronger gains on the harder VTAB-1k Structured split** ($+0.36$ over SA$^2$VP, $+0.97$ over VFPT), the regime where prompt-based methods have historically underperformed.
> > 2. **Generalization across two backbones (ViT and Swin)** with the same mechanism. Table 2 shows our method improves over the next-best Swin baseline by $+0.39 / +0.61 / +1.94$ on Natural/Specialized/Structured, a clearly larger margin than on ViT.
> > 3. **Zero GPU training overhead** for the prior extraction (the Extract-Once strategy; see supplement Sec. 7 and the new main-text Sec. 4.6) and $<2$ ms per-image inference overhead.
> > 4. **Better localization on hard examples**: on the CUB-200 Hard subset (heavy occlusion / complex backgrounds), IoU improves by $+5.7$ over VPT (Table 6 in supplement).
> > 5. **Robustness to operator choice** (Fig. 7): swapping Gabor $\leftrightarrow$ LBP and Sobel $\leftrightarrow$ Canny $\leftrightarrow$ Scharr changes the result by $<0.3\%$, so the gains come from the *presence* of semantic-prior structure rather than from a specific algorithmic choice.
> >
> > We also added a qualitative per-example analysis in supplement Sec. 14, focusing on fine-grained CUB / NABirds / Stanford Dogs images where our method wins by the largest margin (cluttered backgrounds, low-contrast plumage, confusable subspecies).
> >
> > We hope these clarifications and additions address the reviewer's concerns. Thank you again for the constructive feedback.